JCB Journal of Cell Biology

# Calpains orchestrate secretion of annexin-containing microvesicles during membrane repair

Justin Krish Williams[1], Jordan Matthew Ngo[1], Abinayaa Murugupandiyan[1], Dorothy E. Croall[2], H. Criss Hartzell[3], and Randy Schekman[4]

**Microvesicles (MVs) are membrane-enclosed, plasma membrane–derived particles released by cells from all branches of life. MVs have utility as disease biomarkers and may participate in intercellular communication; however, physiological processes that induce their secretion are not known. Here, we isolate and characterize annexin-containing MVs and show that these vesicles are secreted in response to the calcium influx caused by membrane damage. The annexins in these vesicles are cleaved by calpains. After plasma membrane injury, cytoplasmic calcium-bound annexins are rapidly recruited to the plasma membrane and form a scab-like structure at the lesion. In a second phase, recruited annexins are cleaved by calpains-1/2, disabling membrane scabbing. Cleavage promotes annexin secretion within MVs. Our data support a new model of plasma membrane repair, where calpains relax annexin-membrane aggregates in the lesion repair scab, allowing secretion of damaged membrane and annexins as MVs. We anticipate that cells experiencing plasma membrane damage, including muscle and metastatic cancer cells, secrete these MVs at elevated levels.**

## Introduction

Extracellular vesicles (EVs) are membrane-enclosed compartments secreted by cells from all three branches of life. EVs are divided into two subtypes: microvesicles (MVs) that bud directly from the plasma membrane and exosomes that form intracellularly. EVs have utility as biomarkers, as they contain protein and RNA specific to their origin cell. EVs may also function in intercellular communication (Kalluri and LeBleu, 2020). While significant attention has been devoted to exosomes, the mechanisms and physiological circumstances driving MV secretion are poorly understood.

EVs are enriched in annexins (Jeppesen et al., 2019), which bind to phospholipids in the presence of calcium. Annexins have roles in plasma membrane repair. Knockdowns of several individual annexins cause defects in membrane resealing in cultured cells (Koerdt and Gerke, 2017; Sønder et al., 2019). Muscle cells experience high rates of membrane damage in vivo due to repeated contractions and are dependent on the membrane repair machinery for survival. For example, 20% of muscle fibers in rat triceps are disrupted after eccentric exercise (McNeil and Khakee, 1992). Consistent with annexins playing a role in muscle repair, annexin mutations cause muscular dystrophy in mice (Defour et al., 2017; Swaggart et al., 2014). Upon plasma membrane damage, extracellular calcium flows into the cytoplasm. In the presence of micromolar calcium, annexins bind to membranes and may plug lesion sites by tethering membranes (Croissant et al., 2021). These membranes may come from intracellular organelles or from reticulation of the plasma membrane (Croissant et al., 2020). After annexin recruitment, annexin+ MVs are shed from the lesion (Foltz et al., 2021). Recently, Jeppesen et al. separated annexin-rich MVs from exosomes using density gradient centrifugation (Jeppesen et al., 2019). The contents, mechanism of their shedding, and roles of annexin-rich MVs in membrane repair are not understood.

Calpains are cytosolic, calcium-dependent cysteine proteases that, like annexins, have roles in membrane repair. Calpain-1/μ-calpain and calpain-2/m-calpain were the first characterized calpains and are highly abundant across tissues. Calpain-1 and calpain-2 have separate large subunits but share a small subunit. Calpain-3–14 are less abundant and may be restricted to specific tissues (Shapovalov et al., 2022). Like many annexin knockouts (KOs), loss of calpain-1 and/or calpain-2 leads to impaired membrane resealing in cell culture (Prislusky et al., 2024) and severe muscular dystrophy in mice (Piper et al., 2020). Mutations in the muscle-specific calpain-3 produce limb-girdle muscular dystrophy type R1 (Croissant et al., 2021). As with annexins, calpains are activated by micromolar concentrations of calcium in vitro. Because calpain targets include membrane-cytoskeleton anchors, it has been speculated that calpain-1/2

[1]Department of Molecular and Cell Biology, University of California, Berkeley, CA, USA; [2]Department of Biochemistry, Microbiology and Molecular Biology, University of Maine, Orono, ME, USA; [3]Department of Cell Biology, Emory University School of Medicine, Atlanta, GA, USA; [4]Department of Molecular and Cell Biology, Howard Hughes Medical Institute, University of California, Berkeley, CA, USA.

Correspondence to Randy Schekman: schekman@berkeley.edu.

locally detach the plasma membrane from the cytoskeleton around the lesion site (Mellgren et al., 2009). In other systems, calpains also appear to be necessary for MV formation. Platelets release MVs called microparticles during activation and clotting. Microparticle formation in platelets requires sustained elevation of cytosolic calcium and is blocked by calpain inhibitors (Fox et al., 1991). It is unknown if calpains cleave other proteins and facilitate repair and MV formation independently of membrane-cytoskeleton detachment.

Membrane repair appears to be important for cancer cell migration and metastasis (Gounou et al., 2023). Additionally, membrane repair protects cancer cells against T cell–mediated cytotoxicity (Ritter et al., 2022). To characterize membrane repair and mechanisms of MV secretion in cancer cells, we isolated and identified proteins within annexin+ MVs produced by cells grown in culture. After finding several calcium-activated proteins implicated in membrane repair in MVs, we measured annexin secretion in MVs after membrane damage. Addition of sublytic levels of the pore-forming toxin, streptolysin O (SLO), induced an ~20–40-fold increase in vesicular annexin A2 secretion. We found that secreted annexin A1 and annexin A2 are cleaved by calpain-1/2, and that annexins in general are primary targets for calpains. Given that both annexins and calpains are required for membrane repair, we sought to understand the interplay of these proteins during membrane repair. Using an intracellular annexin A2 cleavage reporter, we found that annexin A2 was cleaved after membrane recruitment and wound scabbing, but before secretion. We show that cleaved annexin A2 is deficient in membrane and effector protein binding, and cleavage dissociates annexin A2–linked membranes. Mutant annexin A2 that cannot be cleaved by calpains was recruited normally after membrane damage but was secreted at lower levels in MVs compared with WT annexin A2. Our results link membrane repair to MV secretion and establish a chronology of annexin and calpain function during membrane repair.

## Results

### Annexins and other calcium-responsive proteins are secreted within MVs

To characterize MVs secreted in culture, we utilized serial centrifugation followed by equilibrium density gradient centrifugation. This established method separates EVs based on their buoyant density (Shurtleff et al., 2018). Following a 1,000 (1k) × $g$ centrifugation to remove cells and further centrifugation at 10k × $g$ to remove large EVs, small EVs were sedimented from conditioned medium at 100k × $g$. Sedimented particles were resuspended and placed at the bottom of an iodixanol gradient and centrifuged to separate HCT116-derived small EVs into a low-density, annexin+ population and a high-density, CD63+ population (Fig. 1 A). The CD63+ population is thought to represent exosomes, and the annexin A2+ population is thought to represent a subpopulation of MVs (Jeppesen et al., 2019). Vesicles centrifuged from conditioned medium were immunoprecipitated with CD63 antibody (Fig. 1 B). Known exosome markers, CD9 and alix, coprecipitated with CD63+ vesicles, but annexin A1 did not. CD63+ vesicle proteins were

immunoisolated only when anti-CD63 antibody was pre-conjugated to the beads (Fig. S1 A). We assessed whether annexins were localized to the lumen or the extracellular face of MVs by proteinase K treatment. We found that annexin A2 mostly localized to the lumen of MVs (Fig. 1 C). Luminal flotillin-2 and annexin A2 sedimented with EVs even in the presence of the membrane-impermeable calcium chelator, EGTA, and both were degraded by proteinase K only in the presence of Triton X-100 (TX-100).

Using stable isotope labeling by amino acids in cell culture (SILAC), we identified proteins enriched in the low-density fraction relative to the high-density fraction. EVs secreted by HEK293T cells grown with heavy Arg/Lys or light Arg/Lys amino acids were collected from conditioned medium, centrifuged at 100k × $g$, and pellet material was resuspended and placed beneath a sucrose step gradient (Fig. 1 D). After separation by high-speed centrifugation (Fig. S1 B), heavy amino acid–labeled, low buoyant density EVs, collected from the 10–40% sucrose interface, were mixed with light amino acid–labeled, high buoyant EVs, collected from the middle of the 40% sucrose fraction (and vice versa) prior to mass spectrometry analysis. As expected, conventional exosome proteins (CD63, endosomal-sorting complexes required for transport [ESCRTs], syntenin-1, etc.) were enriched in the high buoyant density fraction (Fig. 1 E and Table S1). Conversely, in the low buoyant density fractions, Ca2+-responsive proteins including annexins, EHD proteins (EHD1 and EHD4), plasma membrane SNARE machinery (SNAP23 and STXBP3), and the Ca2+-transporting ATPase (ATP2B1) were enriched. Annexin, EHD (Demonbreun et al., 2016), and SNARE (Zhen et al., 2021) proteins are all implicated in plasma membrane repair. As expected for plasma membrane–derived vesicles, we also detected several plasma membrane proteins, including CD276 and CD59.

### Membrane damage stimulates secretion of annexin A2+ MVs from the repair site

We showed previously that in addition to exosomes, secretion of this annexin A2+ EV population is stimulated by the Ca2+ ionophore, ionomycin (Williams et al., 2023). In addition, we have previously visualized the secretion of annexin + MVs from muscle cells during Ca2+-dependent membrane repair (Foltz et al., 2021). Given these previous data and our proteomics results, we speculated that secretion of these vesicles might be stimulated by membrane damage. First, we visualized (Fig. 2 A; and Videos 1 and 2) and quantified (Fig. 5 H) FM1-43 staining of HCT116 cells after laser ablation wounding. FM1-43 is membrane impermeable but can stain intracellular membranes brightly if the plasma membrane is disrupted. Within 1 min of ablation, a repair scab of brightly stained membranes formed at the ablation site. For several min after ablation, intracellular membranes slowly stained, suggesting the lesion site was plugged but not fully sealed. As dye influx slowed, vesicles were shed from the damaged cell. These vesicles were intensely stained at a level comparable with the repair site, possibly indicating they were shed directly from the repair site. In addition, as vesicles were shed over the next 5 min, the repair scab slowly lost staining intensity.

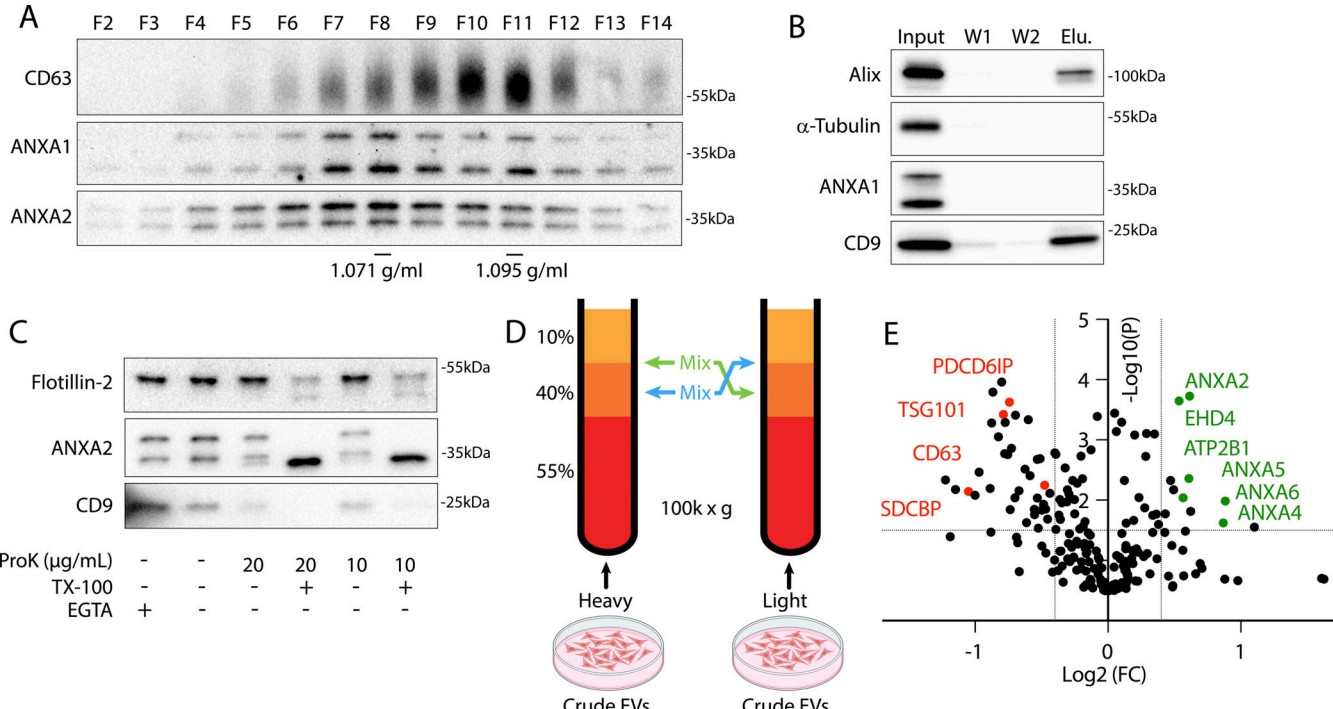

Figure 1. **Annexin-containing EVs are distinct from exosomes. (A)** Immunoblots show the distribution of EV markers across an iodixanol gradient of the conditioned medium 100k × *g* pellet fraction. Samples were taken from low (F2—Fraction #2) to high density (F14—Fraction #14). **(B)** Immunoblots show depletion of annexins from exosomes after immunoprecipitation with anti-CD63 beads from the conditioned medium 100k × *g* pellet fraction (W1—wash #1; W2—wash #2). **(C)** Immunoblots show degradation of EV markers in the conditioned medium 100k × *g* pellet fraction after treatment with indicated combinations of proteinase K (ProK) and 0.1% TX-100. For EGTA treatments, 5 mM EGTA was introduced into the medium prior to centrifuging the conditioned medium at 100k × *g*. **(D)** Schematic illustrating the separation of EVs for quantitative proteomics using sucrose step gradients is shown. **(E)** Volcano plot shows enriched proteins in the high buoyant density fractions (red) relative to the low buoyant density fractions (green). P values were calculated using a *t* test. Source data are available for this figure: SourceData F1.

Next, we visualized annexin dynamics after laser ablation. We focused on annexin A2, as total mRNA sequencing (mRNA-seq) revealed that annexin A2 was likely the most abundant annexin in HCT116 cells (Fig. 2 B). Indeed, there were twice as many annexin A2 reads as the rest of the annexin reads combined. This difference is not due to a difference in transcript size, as annexin A1, A2, A3, and A5 are all ~1.5 kilobases in length. Less than 1 min after ablation, annexin A2 was rapidly recruited to the membrane around the lesion site (Fig. 2 C). For the next 5–10 min after recruitment, annexin A2+ vesicles were shed from the ablation site. As these vesicles were shed, the membrane-recruited annexin A2 dissipated. The morphology of the repair scab varied (Fig. S2 A). Occasionally, annexin A2+ filopodia-like structures emerged rapidly after ablation, before shedding from the damage site. We also observed size variation in annexin A2+ EVs, ranging from 1 μm (Fig. S2 B) to 100–200 nm (Fig. 2 C). In these larger vesicles, we could distinguish complete recruitment of annexin A2 to the membrane, suggesting that the Ca2+ concentration in these vesicles was significantly higher than the intracellular concentration. We also compared repair site recruitment of annexin A2 to annexin A1 (Fig. S2 C) and annexin A6 (Fig. S2 D). More annexin A2 was recruited compared with A1 or A6, further indicating that annexin A2 was the most abundant annexin in HCT116 repair scabs.

To quantitatively assess vesicle shedding during membrane repair, we used the pore-forming toxin SLO to damage the plasma membrane. We began to observe increased cell death at 800 ng/ml SLO as assessed by SYTOX Green staining after treatment (Fig. S2 E). Thus, we considered treatments of 200 ng/ml SLO and less, sublytic. We synchronized pore formation by preincubating cells at 4°C with SLO. At this temperature, SLO binds to the plasma membrane but does not open into a pore (Corrotte et al., 2015). Cells were washed to remove excess SLO, after which pores opened synchronously as the temperature was raised to 37°C.

To assess the dynamics of pore formation using this system, we measured the uptake of SYTOX Green into HCT116 cells in Ca2+-free medium. Because the repair machinery requires Ca2+ influx, under these conditions, SLO pores remained open and were not repaired (Corrotte et al., 2015). SLO pores opened within 40 s to 1 min, as measured by the length of the lag phase in SYTOX Green uptake (Fig. 2 D). Next, we applied SLO to cells expressing a low level of annexin A2-nanoluciferase (Nluc). Using an assay described previously that measures luminal luciferase activity (Williams et al., 2023), we quantified secretion of membrane-enclosed annexin A2-Nluc over time (Fig. 2 E and Fig. S2 F). Neither annexin A2-Nluc nor CD63-Nluc was secreted within the first 2 min of SLO treatment. By 5 min, both annexin A2-Nluc and CD63-Nluc secretion peaked, and by 10 min

Figure 2. **Annexin-containing EVs are shed from the repair scab after plasma membrane damage. (A)** Representative confocal micrographs of FM1-43 infiltration are shown. Image times are relative to the first image taken after ablation. Large arrows in panes I and II indicate the site of ablation. Small arrows in panes II and III indicate the staining of intracellular compartments in an ablated (lower arrows) and a non-ablated cell (upper arrows). Large arrows in panes III and IV indicate EVs. Scale bars: 5 µm. **(B)** Quantification of mRNA-seq reads mapping to each annexin paralog is shown. **(C)** Representative confocal micrographs of ANXA2-mScarlet recruitment after ablation are shown. Image times are relative to the first image taken after ablation. White arrows in panes I and II indicate the site of ablation. Arrow in pane III indicates an EV. Scale bars: 5 µm. **(D)** Sytox Green staining over time in the absence of extracellular $Ca^{2+}$ after cells pretreated with the indicated concentration of SLO was rapidly heated from 4 to 37°C. Error bars indicate three experimental replicates. **(E)** Membrane-protected luminescence in medium fractions at indicated time points after ANXA2-Nluc or CD63-Nluc cells pretreated with 200 ng/µl SLO were rapidly heated from 4 to 37°C is shown. **(F)** EV production index from ANXA2-Nluc or CD63-Nluc cells treated with increasing concentrations of SLO is shown. Error bars indicate three experimental replicates.

secretion had ceased. Accounting for the 1 min it took for SLO pores to open, vesicle secretion initiated 1 min after pore formation and peaked after about 4 min. Next, we measured annexin⁺ MV secretion at increasing concentrations of SLO (Fig. 2 F). Annexin A2-Nluc reached maximal secretion with 100–200 ng/ml SLO, whereas CD63-Nluc, a marker of exosomes, did not reach maximal secretion within the tested concentration range. Annexin A2⁺ MV secretion was stimulated at lower levels of damage, whereas late endosome exocytosis is required at higher levels of damage.

Membrane repair and repair cap shedding involve the phospholipid scramblases, TMEM16F or TMEM16E (Foltz et al., 2021; Wu et al., 2020). We speculated that annexin⁺ EVs may have high levels of anionic phospholipids, such as phosphatidylserine, on the extracellular leaflet if they are derived by

budding from the plasma membrane repair site. We found that annexin A1⁺ and A2⁺ EVs were more efficiently captured by an immobilized form of the phosphatidylserine-binding protein annexin A5 compared with CD63⁺ and Alix⁺ exosomes (Fig. S2 G). Thus, annexin⁺ EVs are enriched in phosphatidylserine and/or phosphatidylethanolamine in their outer membrane leaflet.

Membrane repair also requires ESCRT machinery (Jimenez et al., 2014). Although some reports suggest ESCRTs and annexins work in tandem during membrane repair (Sønder et al., 2019), others have reported independent activity (Jimenez et al., 2014). ESCRT-mediated vesicle fission depends upon the ATPase activity of Vps4. Inducing short-term expression of dominant-negative VSP4a diminished vesicular annexin A2-Nluc signal compared with the uninduced control (Fig. S2 H). We conclude

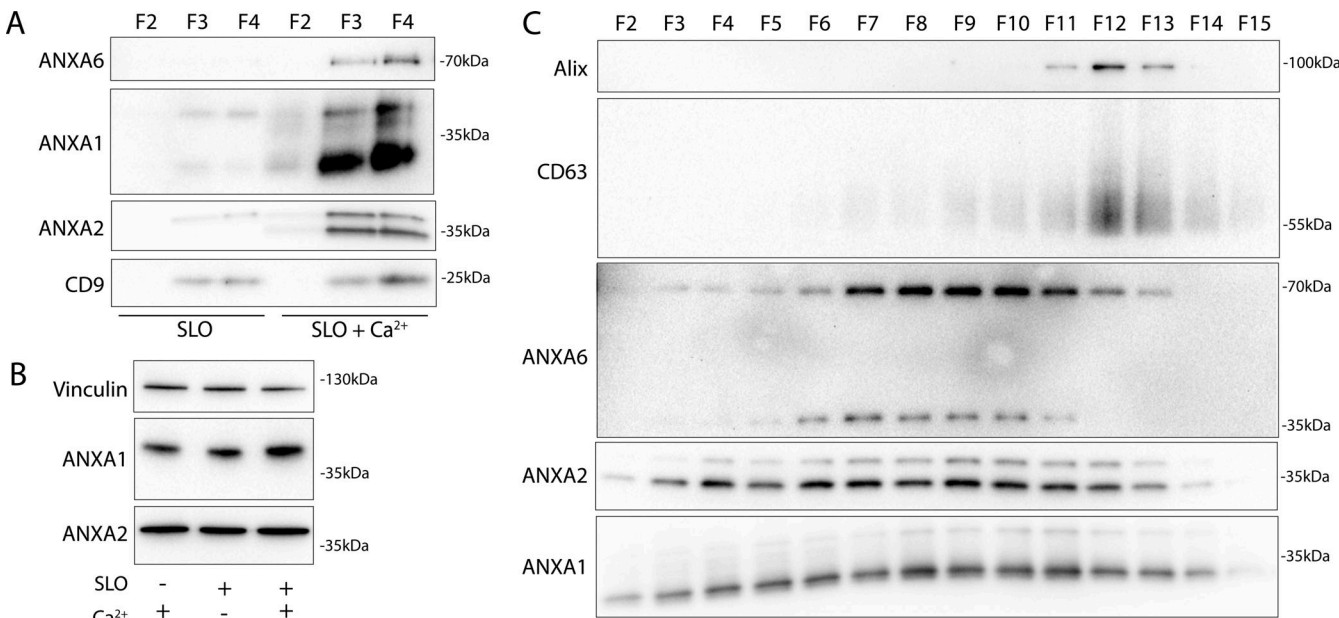

Figure 3. **Annexins within EVs are shifted in apparent molecular weight. (A)** Immunoblots of gradient fractions show enrichment of indicated EV markers after 200 ng/ml SLO treatment, with or without 1 mM Ca²⁺ in the media. F2, F3, and F4 refer to buoyant fractions of a sucrose step gradient of the conditioned medium 100k × *g* pellet fraction. **(B)** Immunoblot analysis of cell lysates after treatment with indicated combinations of 1 mM Ca²⁺ and 200 ng/ml SLO is shown. **(C)** Immunoblots show separation of indicated EV markers after 200 ng/ml SLO treatment. F2–F15 refer to fractions #2–15 of an iodixanol gradient of the conditioned medium 100k × *g* pellet fraction, moving from low to high density. Source data are available for this figure: SourceData F3.

that annexin⁺ MV shedding during membrane repair at least partially depends on the ESCRT pathway.

**Annexins within MVs have altered apparent molecular weight**
Next, we examined the role of extracellular Ca²⁺ in the secretion of these annexin⁺ vesicles. Significant levels of annexin proteins were detected in buoyant fractions of a sucrose density gradient, but only when Ca²⁺ was present in the medium (Fig. 3 A). We noted that annexin A1 and A2 within MVs migrated at a lower apparent molecular weight. No such annexin mobility shifts were detected in cell lysates even after SLO treatment (Fig. 3 B). We confirmed that annexin proteins were secreted in a distinct vesicle fraction from exosomes after SLO treatment. As expected, annexin A1, A2, and A6 were present in a low-density fraction relative to high-density CD63⁺ exosomes (Fig. 3 C). Annexin A6 also appeared to shift in molecular weight, although the apparent molecular weight change was much larger for annexin A6 (~70 to ~35 kDa) than for annexin A1 (~38 to ~34 kDa) or A2 (~37 to ~35 kDa). Because muscle cells experience some of the highest rates of membrane damage, we examined this effect using C2C12 cells. As with tumor cells, myotubes, differentiated from C2C12 cells, secreted EVs with apparently processed annexins in response to SLO treatment (Fig. S3 A).

We speculated that SLO pores may be present on annexin⁺ vesicles derived from the secreted plasma membrane repair scab. Indeed, others reported such pores on vesicles secreted from SLO-treated cells (Romero et al., 2017). In our experiments, SLO was secreted predominantly on larger vesicles that sedimented at 10k × *g* (Fig. S3 B). We reasoned that such perforated vesicles would be accessible to membrane-impermeable

compounds. As expected, a larger fraction of annexin A2-Nluc secreted in larger vesicles was accessible to a membrane-impermeable Nluc inhibitor compared with annexin A2-Nluc secreted in smaller vesicles (Fig. S3 C). We conclude that cells secrete both perforated and sealed EVs in response to membrane damage.

**Calpain-1/2 proteases cleave annexins during MV shedding**
We investigated the basis of the apparent molecular weight change of the annexins in MVs. Annexins are substrates for phosphorylation by kinases including Src or PKC (Bharadwaj et al., 2013) and for proteolysis by enzymes including metalloproteases (Blume et al., 2012), plasmin, cathepsins, and calpains (Williams et al., 2010). We tested the role of Ca²⁺ in the apparent molecular weight change for annexins and found that the addition of 1 mM in a lysate of HCT116 cells resulted in a rapid conversion of annexin A2 (Fig. 4 A).

Calpains are a well-characterized family of Ca²⁺-dependent proteases that are necessary for membrane repair in cells. Calpains 1 and 2 are the highest expressed calpains across tissues and have ~10 fold higher transcript levels in HCT116 cells compared with other calpains (Jin et al., 2023). Addition of the highly specific calpain 1 and 2 inhibitor, calpastatin domain I ($K_i$ = 15 nM), inhibited the apparent molecular weight shift at a concentration near the $K_i$ (Fig. 4 B). The less specific, membrane-permeable calpain inhibitor, ALLN ($K_i$ = 200 nM), also inhibited the apparent molecular weight shift near the $K_i$ (Fig. S4 A). We purified annexin A2-halotag using a one-step, tag-free purification method (Fig. S4 B). Annexin A2-halotag gel mobility was shifted in apparent molecular weight from ~70 to ~67 kDa when combined with purified calpain-1 and Ca²⁺

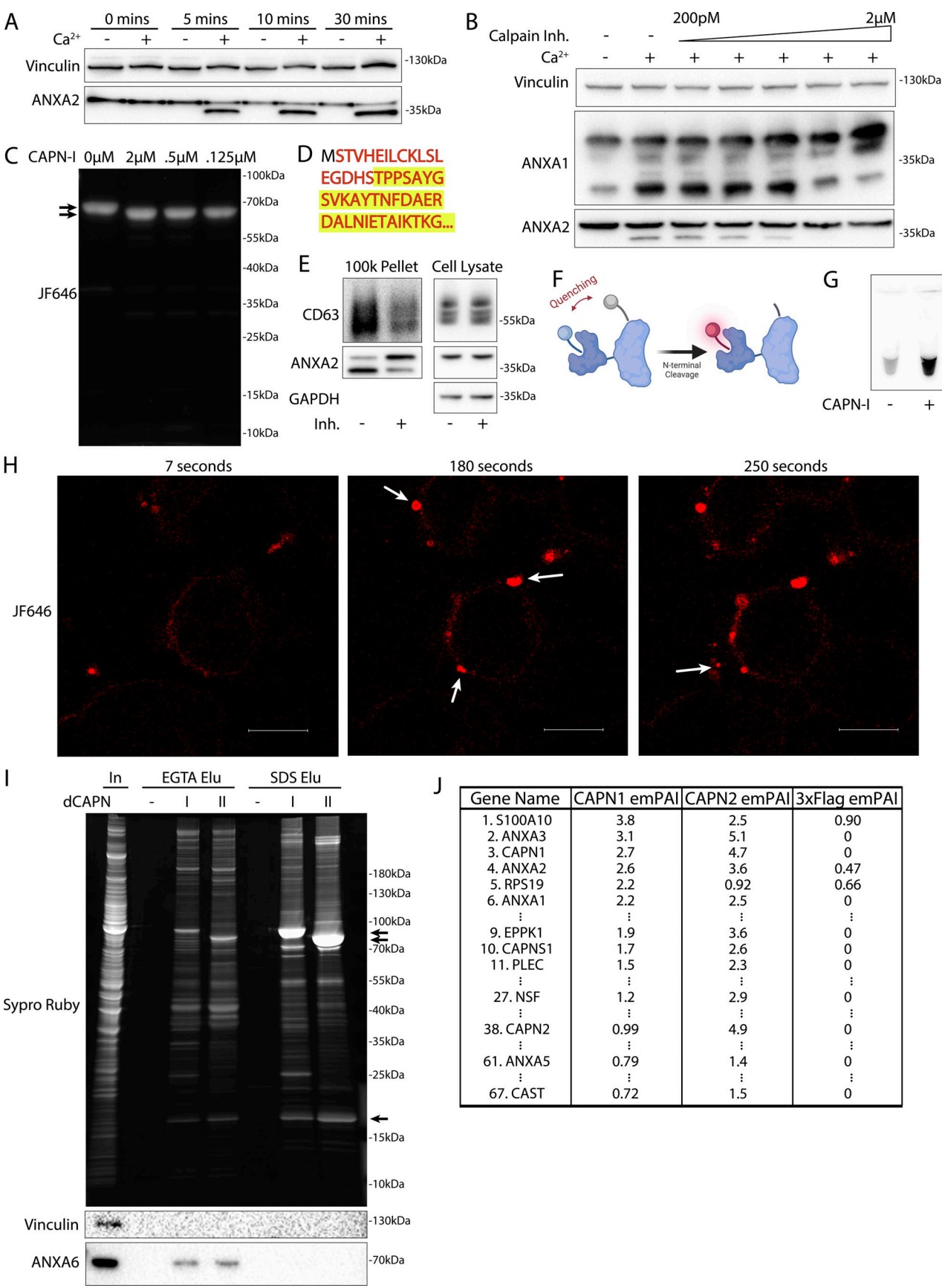

Figure 4. **Calpain-1/2 cleaves annexins, which are then shed in MVs. (A)** Immunoblot analysis of cytosol fractions after incubation with or without 1 mM Ca$^{2+}$. **(B)** Immunoblot analysis of cytosol fractions after incubation with or without 1 mM Ca$^{2+}$, with a range of concentrations of calpastatin domain I inhibitor

(Calpain Inh.). **(C)** In gel fluorescence of JF646-labeled, recombinant annexin A2-Halo incubated with indicated concentration of purified, porcine calpain-1. Arrows indicate uncleaved and cleaved products. **(D)** Mapping of tryptic peptides to the first 50 amino acids of annexin A2. Mass spectrometry analysis of recombinant annexin A2 (red text) is compared with tryptic digest-mass spectrometry analysis of recombinant annexin A2 treated with purified, porcine calpain-1 (yellow highlight). **(E)** Immunoblot analysis of cell lysate and conditioned medium 100k × *g* pellet fraction after treating cells with 200 ng/ml SLO in the presence or absence of 10 μM calpain inhibitor, ALLN. **(F)** Schematic illustrating the recombinant annexin A2-Halo reporter, labeled with a 5WS maleimide quencher on the N terminus and a JF646 halo ligand on the C terminus. **(G)** JF646 fluorescence of an in vitro reaction containing 5 μM self-quenched annexin A2 with or without 0.5 μM porcine calpain-1. **(H)** Representative confocal micrographs of dequenched annexin A2-Halo-JF646 fluorescence in cells. Times are relative to the first image taken after addition of 1 mM $Ca^{2+}$. White arrows in pane II indicate puncta on the cell periphery. Arrow in pane III indicates an EV. Scale bars: 10 μm. **(I)** Total protein (Sypro Ruby staining) and immunoblot analysis of the substrate-trapping experiment, using 3xFlag (−), 3x-Flag C115S calpain-1 (dCAPN-I), or 3x-Flag C105S calpain-2 (dCAPN-II) as bait, is shown. Arrows indicate calpain proteins. **(J)** Table listing proteins identified in EGTA elutions from 3xFlag, 3x-Flag C115S calpain-1 (dCAPN-I), or 3x-Flag C105S calpain-2 (dCAPN-II) capture experiments. Proteins are listed by exponentially modified protein abundance index (emPAI) in the dCAPN-I elution. Keratin proteins were not included in the list. Source data are available for this figure: SourceData F4.

(Fig. 4 C). The cleavage appeared to be remarkably specific, producing only the 67-kDa species, even in incubations at equal concentrations of calpain and annexin A2-halotag (2 μM). Purified calpain-1 also cleaved recombinant annexin A6 into 35-kDa products (Fig. S4, C and D), which matched the shift in apparent molecular weight of annexin A6 in MVs (Fig. 3 C). The consensus sequence that predicts the substrate specificity of calpain-1/2 is not well-defined (Sorimachi et al., 2012). Therefore, we mapped the calpain cleavage site on annexin A2 by mass spectrometry and detected peptides covering the first 50 amino acids of the protein (Fig. 4 D and Table S2). Mass spectrometry of processed annexin A2 revealed a cleavage that removed the first 18, resulting in a polypeptide starting at residue Thr-19. The conversion of annexin A2 in MVs was blocked in cells treated with ALLN (Fig. 4 E). We conclude that MV annexins are cleaved by $Ca^{2+}$-dependent calpain-1/2.

Given that both annexins and calpains are important for membrane repair, we sought to understand the interplay between these two proteins. Based on the location of the cleavage site (Ser-18/Thr-19), we designed a reporter for annexin A2 cleavage by calpain (Fig. 4 F). Purified annexin A2-halotag was labeled with a fluorescent halo ligand and a maleimide fluorescence quencher. As the only accessible cysteine on annexin A2-halotag is Cys8 (Madureira et al., 2011; Samelson et al., 2018), the cleavable N terminus could be site specifically labeled with the quencher. The addition of calpain-1 to the reporter led to a robust increase in fluorescence in vitro as the N terminus containing the quencher was cleaved (Fig. 4 G). Next, we used this reporter in cells, using a previously described delivery protocol (Teng et al., 2018; Walev et al., 2001). Cells were treated with SLO and the reporter in the absence of calcium, and the reporter was allowed to diffuse into the cells. After washing the cells, the addition of calcium triggered the repair process, and annexin A2 cleavage was visualized in the cells as an increase in halotag-JF646 fluorescence (Fig. 4 H). Within 1–2 min, bright, cleaved annexin A2-halotag puncta formed on distended parts of the cell surface, presumably at repair scabs. After puncta formation, vesicles could be seen ejecting from these puncta. These vesicles were also brightly stained, similar to the repair scab puncta. Thus, annexin A2 cleavage by calpain is highly localized in cells and occurs after annexin A2 recruitment and scabbing at the plasma membrane.

Many proteins are reported substrates for calpains. We wondered if annexin proteins are primary targets for calpains or if they represent a minor fraction of total. We developed an unbiased, substrate-trapping approach to identify calpain-1/2 substrates (Fig. S4 E). Calpains (3xFlag-tagged), inactivated by mutation of the catalytic cysteine to serine, were immobilized on anti-flag beads. Cytosol was incubated with the beads and substrates in the presence of $Ca^{2+}$ bound to the open calpain active site. Addition of EGTA-chelated $Ca^{2+}$ to close the active site, which resulted in substrates eluting from the beads. Stable, $Ca^{2+}$-independent interactors were retained. We coexpressed and purified CAPN1[C115S]-3xFlag-6xHis (dCAPN-I) and CAPN2[C105S]-3xFlag-6xHis (dCAPN-II), each in complex with calpain small subunit, CAPNS1(86–268) (Fig. S4 F).

Using dCAPN-I and dCAPN-II baits, we captured calpain substrates (EGTA elution) and stable interactors (subsequent SDS elution) from HCT116 cytosol (Fig. 4 I). Consistent with calpains targeting annexins in the presence of $Ca^{2+}$, EGTA quantitatively eluted annexin A6 from dCAPN-I and dCAPN-II baits. Many known calpain substrates are targeted by both calpain-1 and calpain-2. Consistent with these observations, the gel patterns of the dCAPN-I and dCAPN-II EGTA elution were similar. Next, we analyzed EGTA elutions using high-resolution mass spectrometry and ranked proteins by their exponentially modified protein abundance index in the dCAPN-I sample (Fig. 4 J and Table S3). As expected, the endogenous calpain-1/2 inhibitor, calpastatin, was captured by dCAPN-I and dCAPN-II baits, but not by the 3xFlag alone control. In addition, we recovered plectin (PLEC), a known calpain substrate that links the cytoskeleton to the plasma membrane (Muenchbach et al., 1998). The significant abundance of PLEC and EPPK1 exclusively in the dCAPN-I and dCAPN-II elutions is consistent with the role suggested for calpains in severing the plasma membrane from the cytoskeleton.

S100A10, annexin A3, annexin A2, and annexin A1 were the top few substrates in the dCAPN-I sample. In addition, we found six other annexins (A4, A5, A6, A7, and A11) with lower peptide counts. High levels of all the same annexins were identified in the dCAPN-II sample, but not in the 3xFlag alone control. We also found many novel potential calpain-1/calpain-2 substrates, including NSF and AHNAK, the latter of which forms a complex with annexin A2 and S100A10 that may be important for plasma membrane repair (Rezvanpour et al., 2011). Our data suggest

that annexins and potentially annexin interactors during membrane repair are a major target for calpain-1/2.

## Calpain cleavage decreases annexin A2's membrane binding and scabbing activity

We next considered how cleavage changes annexin behavior at the repair site. To investigate this, we designed two annexin A2 mutants. The published structure of calpain-2 with its endogenous inhibitor, calpastatin, suggests two calpastatin prolines are important for recognition (Hanna et al., 2008). Annexin A2 also has two prolines (Pro-20 and Pro-21) directly adjacent to the calpain cleavage site. Mutating this site (P20D/P21D) (mt-annexin A2) rendered calpain incapable of cleaving annexin A2 (Fig. 5 A). We also created a second truncation mutant, annexin A2(18–339) (tr-annexin A2), that reproduced annexin A2 after cleavage (Fig. 5 B). For technical reasons, SLO was added to cells at 37°C. This procedure resulted in delayed, asynchronous pore formation and repair.

Although cleaved annexin A2 retained some Ca²⁺-dependent membrane association activity in vitro (Fig. S5 A), its membrane binding was attenuated (Fig. 5 C). High doses of SLO cause large scale translocation of annexins to the plasma membrane (Babiychuk et al., 2009). After HCT116 cells were treated with a high, near lytic (400 ng/ml) dose of SLO, tr-annexin A2-mNeonGreen was recruited more slowly to the plasma membrane compared with WT-annexin A2-mScarlet. In addition, a significant pool of tr-annexin A2 remained cytosolic, unlike WT-annexin A2, which was almost completely recruited to the membrane. Mt-annexin A2 was recruited as quickly as WT-annexin A2 to the plasma membrane (Fig. S5 B).

S100 proteins bind to and modify annexin function. The first 10–12-amino acid segment of annexin A2 binds to S100A10 (Kd = 13 nM) (Réty et al., 1999). Thus, we predicted that calpain-cleaved annexin A2 would be incapable of interacting with S100A10. Immunoprecipitation of WT- and mt-annexin A2-HA captured S100A10. As expected, however, tr-annexin A2-HA no longer coprecipitated S100A10 (Fig. 5 D). Because previous reports suggest that annexin A2 may interact with other S100 proteins (Jaiswal et al., 2014), we tested the specificity of the annexin A2–S100A10 interaction. Using annexin A2-Halo or calpain-treated annexin A2-Halo as bait, we captured annexin A2–interacting proteins (Fig. S5 C). Two proteins of apparent molecular masses of 10 and 40 kDa appeared in the annexin A2-halo elution but not in the calpain-treated annexin A2-halo elution. Gel excision-mass spectrometry identified these proteins as annexin A2 and S100A10, with no peptides mapping to any other S100 protein.

Annexin A2 and S100A10 formed an oligomer (Fig. S5 F). We wished to test the possibility that this oligomer promoted the tight recruitment and membrane-bridging activity of annexin A2. A reduced binding of S100A10 might account for the lowered membrane recruitment of tr-annexin A2. The membrane recruitment of S100A10 was enhanced in comparison with WT-annexin A2 after HCT116 cells were treated with a high dose of SLO (Fig. 5 E). We noted that S100A10 required annexin to bind membranes (Fig. S5 D) and was destabilized and degraded in vivo after annexin A2 depletion (Bharadwaj et al., 2021). This

suggested that all membrane recruited S100A10 is bound to a pool of annexin A2 and that calpain dissociates the S100A10-annexin A2 dimer of dimers.

We confirmed that annexin A2 is required for membrane repair in our HCT116 cell model. Using a CRISPR Cas9 intron trapping approach (Reber et al., 2018), we generated an annexin A2 KO (Fig. 6 B). Scabbing/plugging was monitored over time in the presence of a high concentration of Sytox Green (2.5 μM) and SLO. After 6 min of SLO treatment, significantly more Sytox Green entered annexin A2 KO cells compared with WT HCT116 cells (Fig. 5 F). This difference was dependent on the addition of calcium to the medium (Fig. S5 E). In a 37°C incubation, SLO binding and pore formation took closer to 3–4 min (Fig. S5 E), suggesting that the increased Sytox Green influx in annexin A2 KO cells occurred within 2 min of pore formation. We next tested the effect of annexin A2 KO on repair scab formation (Fig. 5, G and H). WT cells ablated with a laser in the presence of FM1-43 revealed a brightly stained repair scab distended from the ablation site by 1 min after injury. This repair scab was nearly absent in annexin A2 KO cells. We conclude that annexin A2 KO caused a defect in scabbing during the early stages of membrane repair.

Annexin A2-S100A10 dimer of dimers aggregate liposomes in vitro (Drücker et al., 2013) in a manner that may resemble lesion site scabbing in vivo. We purified both annexin A2 and annexin A2-S100A10 dimer of dimers (Fig. S5 F) and mixed them with liposomes. Addition of annexin A2 or, to a larger extent, addition of annexin A2-S100A10 caused liposomes to aggregate (Fig. 5, I and J). Addition of calpain-1 dissolved annexin A2 and annexin A2-S100A10 liposome aggregates. Thus, calpain cleavage reduces annexin A2 membrane recruitment, terminates annexin A2-S100A10 association, and dissolves annexin A2–mediated membrane scabs.

The catalytic activity of the calpains is upregulated by PI(4,5)P₂, allowing the proteins to partially associate with membranes in vivo (Leloup et al., 2010). We wondered whether calpains, like annexins, could be recruited to membranes in the presence of calcium (Fig. S5 G). We centrifuged membranes from the post-nuclear supernatant (PNS) of a cell lysate, either in the presence or absence of 1 mM Ca²⁺. Membranes sedimented in the presence of Ca²⁺ were washed with 5 mM EGTA. High levels of annexin A1 and A2 and activated, autolyzed calpain-1 were eluted and recovered in the wash supernatant sample. In contrast, annexins and calpain-1 did not sediment along with membranes when 1 mM Ca²⁺ was not added to the PNS. Thus, calpains are also recruited to membranes in the presence of calcium Ca²⁺, either by directly binding membrane lipids or by interacting with membrane-associated proteins.

## Inhibiting annexin cleavage causes defects in repair scab secretion during membrane repair

We next tested the role of calpain-mediated cleavage in the shedding of annexin A2 scabs (Fig. 6 A). Calpain inhibition significantly lowered annexin A2⁺ MV secretion. To test the specificity of the effect on annexin A2, we transfected annexin A2 KO cells with WT-annexin A2-Nluc or mt-annexin A2-Nluc (Fig. 6 b). SLO-treated cells secreted lower levels of vesicular

Figure 5. **Calpain cleavage attenuates the membrane binding and scabbing activity of annexin A2. (A)** Immunoblot analysis of lysates from cells expressing WT ANXA2-HA or ANXA2[P20D, P21D]-HA. $Ca^{2+}$ (1 mM) was added to lysis buffer where indicated. **(B)** Immunoblot analysis of lysates from cells expressing WT ANXA2-HA or ANXA2-HA with the indicated N-terminal truncations. $Ca^{2+}$ (1 mM) was added to lysis buffer where indicated. **(C)** Representative confocal micrographs of ANXA2-mScarlet (wt-mScarlet) and ANXA2(18–339)-mNeonGreen (Tr-mNeonGreen)–expressing cells. Image times are relative to the addition of 400 ng/ml SLO. Scale bars: 5 µm. **(D)** Immunoblot analysis of lysates and anti-HA immunoprecipitation elutions from cells expressing WT ANXA2-HA (wt-HA), ANXA2(18-339)-HA (Tr-HA), ANXA2[P20D, P21D]-HA (Mt-HA), or no HA construct. **(E)** Representative confocal micrographs of ANXA2-mScarlet (wt-mScarlet) and S100A10-mNeonGreen–expressing cells. Image times are relative to the addition of 400 ng/ml SLO. Arrows indicate cells with annexin A2 and S100A10 translocating to the membrane. Scale bars: 5 µm. **(F)** Representative widefield micrographs of WT or annexin A2 KO cells, with 2.5 µM Sytox Green added for 6 min. SLO (200 ng/ml) was added with Sytox Green in the indicated panels. Scale bars: 100 µm. **(G)** Representative confocal micrographs of FM1-43–stained (2.5 µM) WT or annexin A2 KO cells, 100 s after laser ablation. White arrows indicate the ablation site. Scale bars: 5 µm. **(H)** Quantification over time of the repair scab intensity from FM1-43–stained (2.5 µM) WT or annexin A2 KO cells. Error bars indicate six experimental replicates. **(I and J)** Representative widefield micrographs (I) and aggregate size quantification (J) of Texas Red-labeled 200-nm liposomes mixed with the indicated combinations of 300-nM annexin A2 (A2), annexin A2-S100A10 (A2+S1), or Calpain-1 (C1). Scale bars: 50 µm. Error bars indicate two experimental replicates. Source data are available for this figure: SourceData F5.

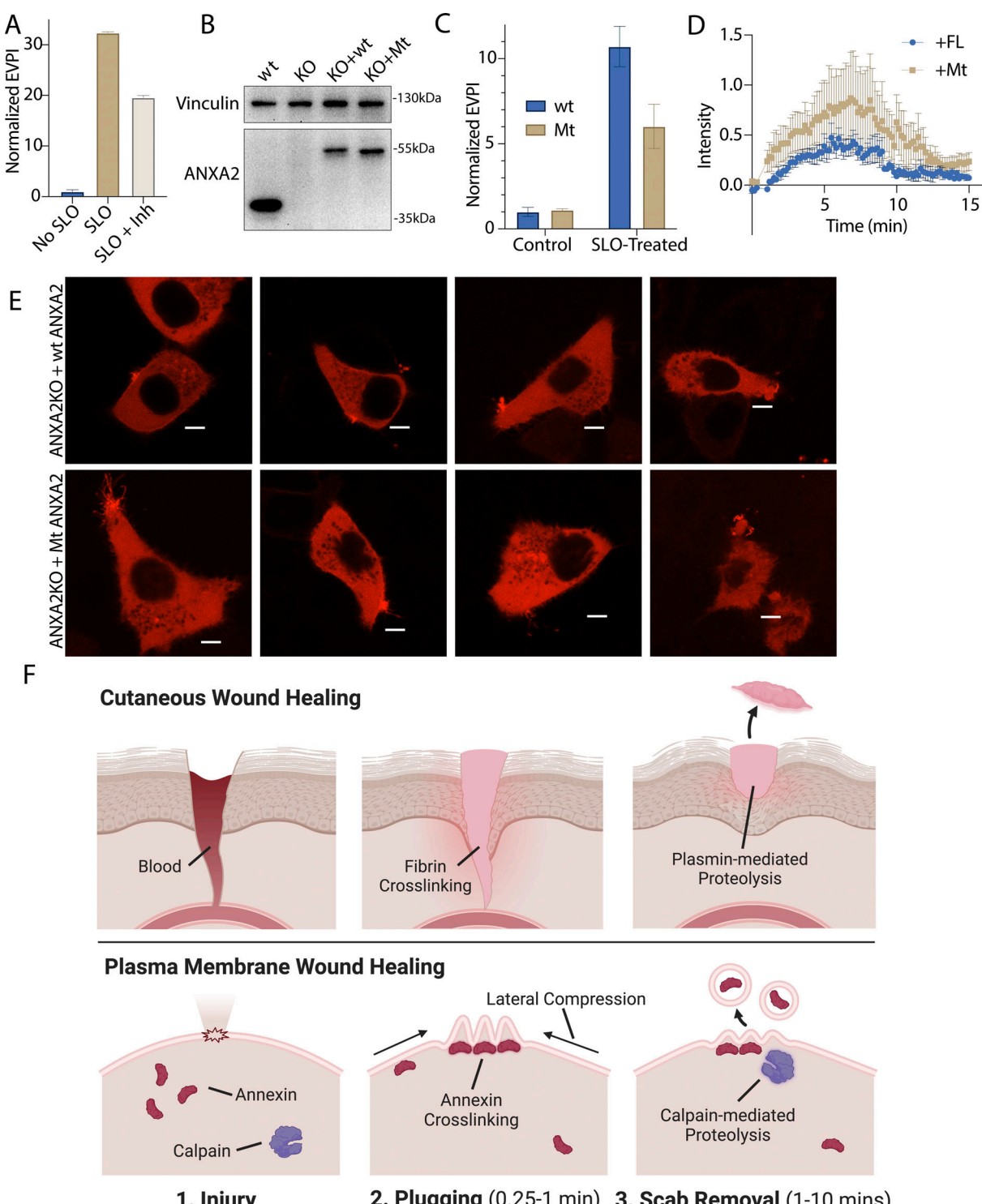

Figure 6. **Inhibition of annexin A2 cleavage decreases annexin A2⁺ MV secretion during membrane repair. (A)** EV production index from ANXA2-Nluc cells treated with the indicated combinations of 200 ng/ml SLO and 20 µM calpain inhibitor, ALLN. Error bars indicate three experimental replicates. **(B)** Immunoblot analysis of lysates from WT cells (wt), annexin A2 KO cells, and annexin A2 KO cells expressing WT annexin A2-Nluc (KO + WT) or ANXA2 [P20D, P21D]-Nluc (KO + Mt). **(C)** EV production index from WT ANXA2-Nluc (wt) or ANXA2[P20D, P21D]-Nluc (Mt) cells treated with or without SLO. Error bars indicate three experimental replicates. **(D)** Quantification over time of the repair scab intensity from FM1-43–stained (2.5 µM) annexin A2 KO cells rescued with ANXA2-mScarlet (wt) or ANXA2[P20D, P21D]-mScarlet (Mt). Error bars indicate six experimental replicates. **(E)** Confocal micrographs of eight laser-ablated annexin A2 KO cells rescued with either WT ANXA2-mScarlet (wt) or ANXA2[P20D, P21D]-mScarlet (Mt). Images are 2 min, 30 s after ablation. Scale bars: 5 µm. **(F)** Schematic depicting the current model of plasma membrane repair and annexin⁺ MV secretion. EPVI, EV production index. Source data are available for this figure: SourceData F6.

mt-annexin A2 compared with WT-annexin A2 (Fig. 6 C). Next, we transfected annexin A2 KO cells with WT-annexin A2-mScarlet or mt-annexin A2-mScarlet and imaged rescue after laser ablation (Fig. 6, D and E). Transfection of annexin A2-mScarlet partially rescued repair scab formation. Quantification of the FM1-43–stained repair scab revealed that cells rescued with mt-annexin A2-mScarlet had a larger repair scab compared with cells rescued with WT-annexin A2-mScarlet, with some overlap in the error bars during the time course (Fig. 6 D). Visualization of annexin A2-mScarlet after ablation revealed that the mt-annexin A2+ repair scab often had long (5 µm) filopodia-like structures extending from the repair site. In contrast, the mt-annexin A2+ repair scab was smaller and more compact. We conclude annexin cleavage is an integral part of the membrane repair and shedding process.

## Discussion

### Annexin-dependent membrane repair is a two-step process
We find that calpains target annexins during plasma membrane repair, driving the secretion of damaged membranes as MVs, and propose a model of plasma membrane repair that is analogous to cutaneous wound repair (Fig. 6 F). After epidermal damage, blood flows into the wound and clots. Clotting is driven by fibrin cross-linking and scabbing over the wound. In a second, slower process, proteases including plasmin and collagenase cut the cross-links, and the scab eventually falls off the repaired wound (Kearney et al., 2022). During plasma membrane repair, annexins, including annexin A2, are rapidly recruited to the plasma membrane within 45 s of damage induced by laser ablation (Fig. 2 C) or SLO (Fig. 5 C). An annexin+ membrane scab-like structure forms at the lesion site that may be visualized with the dye, FM1-43, and slows FM1-43 staining of intracellular organelles after laser ablation (Fig. 2 A). These "scabs" have either condensed or filipodia-like structures (Fig. S2 A). Because the cell simultaneously decreases in size (Videos 1 and 2), we hypothesize that the scab is derived from lateral compression of the plasma membrane. Like fibrin, annexin A2 and other annexins may mediate this scabbing by cross-linking membranes at the wound site. In vitro, annexins cross-link liposomes (Fig. 5, I and J). Annexin A2 KO cells lose latency to a membrane-impermeable dye within 2 min of wounding with SLO (Fig. 5 F), a period that coincides with annexin A2 recruitment to the membrane lesion (Fig. 2 C). Finally, the FM1-43–stained repair scab is virtually absent in laser-damage annexin A2 KO cells (Fig. 5, G and H). S100 proteins may promote this process by enhancing annexin recruitment (Fig. 5 E) and scabbing (Fig. 5, I and J). In a second stage, as for plasmin, calpain-1/2 cleaves annexins (Fig. 3, A and C; and Fig. 4, A–E). Calpain cleavage of annexin A2 peaks by 1–2 min and is spatially restricted within putative repair scabs (Fig. 4 H). Calpain cleavage of annexin A2 dissociates aggregated liposomes in vitro (Fig. 5, I and J) and prevents S100A10 binding (Fig. 5 D). Calpain cleavage and dissociation promote the secretion of annexin A2 in MVs after SLO wounding (Fig. 6, A–C). However, calpain inhibition only partially blocks SLO-induced shedding, indicating other mechanisms may drive shedding from the repair site.

Secretion of both perforated and closed vesicles (Fig. S3, B and C) peaks by 5 min and abates by 10 min (Fig. 2 E). Rescue of annexin A2 KO with uncleavable annexin A2 causes a larger repair scab to form after laser wounding (Fig. 6, D and E). This descabbing process may increase the flexibility of the repair scab to facilitate budding and shedding, leading to a smaller repair scab at steady state. Because of their ability to induce strong membrane curvature, annexins form blebs (Boye et al., 2018) and lattice-like sheets in vitro (Lin et al., 2020). Cleavage may destabilize these structures, potentially preventing annexin overactivation.

Depending on the type of plasma membrane wound and the extent of damage, different repair machinery and mechanisms may be employed. Although we and others have visualized ectocytosis at the membrane damage site (Foltz et al., 2021; Wu et al., 2020), other mechanisms, including endocytosis of damaged membranes, are also reported (Stefani et al., 2024; Zhen et al., 2021). Additionally, we previously showed that late endosomes fuse with the plasma membrane during membrane repair (Williams et al., 2023), and others have observed early endosomes (Raj et al., 2023) and lysosomes (Reddy et al., 2001) fusing with the plasma membrane during membrane repair. Here, we observe that late endosome exocytosis, measured by CD63+ exosome secretion, appears to be reserved for higher levels of plasma membrane damage compared with ANXA2+ MV secretion (Fig. 2 F). We hypothesize that these diverse repair mechanisms may occur in concert to heal a plasma membrane lesion.

Whereas some reports suggest that calpains and annexins are important for pore toxin repair (Prislusky et al., 2024), other reports propose that this machinery is reserved for repair of larger lesions (Jimenez et al., 2014; Piper et al., 2020). Although our data support that annexins and calpains are involved in repairing smaller SLO pores, this machinery may be even more important for repair of large lesions. Compared with SLO wounding, laser ablation likely generates larger wounds and may affect integrity of the actin cytoskeleton proximal to the wound. We advise caution when directly comparing conclusions from SLO and laser-wounding experiments.

### Substrate trapping reveals new targets for calpain-1/2
Using unbiased proteomics, we identified targets for calpain-1 and -2. Because calpain substrate recognition appears not to rely on primary sequence alone, computationally predicting calpain targets is challenging. Our dataset predicts hundreds of new calpain targets, which may aid the development of new prediction algorithms. Although calpains were known to cleave membrane-cytoskeleton anchors during repair (Mellgren et al., 2007), we have extended the targets to include annexins A1, A2, and A6 (Fig. 4, B, I, and J; and Fig. S4 D). However, we do not exclude membrane-cytoskeleton anchor proteins as important calpain targets. When preparing the input lysate for substrate trapping, we depleted membranes and organelles by centrifuging the lysate at high speed ($\sim$100k × g). This step prevented pulldown of membranes and associated, nonspecific membrane proteins, as calpains interact with membranes in the presence of calcium (Fig. S5 G). However, high-speed ultracentrifugation

may have decreased membrane-cytoskeleton anchor protein abundance in the input lysate.

A comparison of the annexin transcripts suggest that annexin A2 may be the most abundant annexin in HCT116 cells (Fig. 2 B), possibly explaining why annexin A2 KO HCT116 cells are defective in membrane scabbing (Fig. 5, F–H). In other cell lines, however, other annexins may be more abundant. For all tested annexins (A1, A2, and A6), calpain cleavage is predicted to terminate membrane-bridging activity. For the single annexin domain proteins, like annexin A2, calpains cleave at the N terminus (Fig. 5 D), preventing dimerization. For annexin A6, a tandem annexin domain protein, calpain cleaves between the annexin domains (Fig. 3 C and Fig. S4 D). Thus, no matter the annexin, calpain cleavage should terminate the membrane-bridging activity of the annexin, potentially promoting scab breakdown. Besides annexin A1, A2, and A6, our substrate-trapping experiment suggests many other annexins are also targeted by calpains (Fig. 4, I and J).

Annexin A2 is thought to form a "membrane repair complex" with S100A10, AHNAK, and with dysferlin in muscle cells (Rezvanpour et al., 2011). This complex may mediate calcium-dependent patching of membrane lesions. AHNAK and S100A10 were identified in our proteomic analysis of calpain targets. In muscle cells, dysferlin interacts with annexin A2 and is also directly targeted by calpain 1/2 (Redpath et al., 2014). We suggest that by targeting annexins, AHNAK, and dysferlin, calpains may control membrane-scabbing proteins during calcium-dependent membrane repair. Low-level expression of a C-terminal truncation mutant of annexin A6, similar to the calpain-cleaved form of annexin A6, caused dominant-negative inhibition of membrane repair (Demonbreun et al., 2016; Swaggart et al., 2014). This observation supports our view that annexin cleavage destabilizes the repair cap and likely occurs after scabbing. If calpains cleave annexin A6 prior to annexin recruitment, scabbing/repair cap formation is inhibited.

Calpain cleavage may modulate aspects of annexin function other than cell surface scabbing. The N-terminal domain of annexins interacts with other proteins directly or indirectly through S100 proteins. Annexin A2, for example, may interact with the cytoskeleton (Jaiswal et al., 2014; Prislusky et al., 2024). Calpain cleavage would terminate such an interaction. In this model, calpains-1/2 would act in accordance with their previously described function, targeting connections between the actin cytoskeleton and the plasma membrane (Dourdin et al., 2001; Mellgren et al., 2007). Both calpain activities, the descabbing of annexins, and the detaching of the membrane from the cytoskeleton may be important for membrane repair and shedding.

### Ca²⁺-dependent production of MVs secretes annexins and annexin cleavage products

Lower levels of annexin-containing MVs are secreted into the culture medium by unstimulated cells (Fig. 1, A–C). It is tempting to speculate that some low level of membrane damage, repair, and MV secretion occurs continuously. Consistent with this model, MVs are enriched with an array of $Ca^{2+}$-responsive proteins (Fig. 1 E), many of which are important for membrane

repair. Also, the annexins in constitutively secreted MVs are cleaved by calpains, which require micromolar calcium influx for activation. Such "damage" may be a consequence of the passaging of cells or the use of bovine serum, which contains complement proteins capable of perforating human cells (Triglia and Linscott, 1980). Other processes such as cell death or the activation of a very high flux channel (e.g., P2X7) may elevate $Ca^{2+}$ to levels sufficient for the secretion of these vesicles (Golia et al., 2023). We suggest that these are normal processes that occur in animals and may explain the representation of MVs in all bodily fluids.

N-terminal peptides from annexins, particularly from annexin A1, mediate anti-inflammatory signaling. Annexin A1–derived peptides bind to formyl peptide receptors on the surface of immune cells, dampening proinflammatory signals and promoting proresolving processes such as efferocytosis (Perretti and Dalli, 2023). We propose that calpains may also generate these peptides. Once cleaved, N-terminal peptides may diffuse directly through the membrane or through vesicular disruptions. Annexin A1 N-terminal peptide intercalates into phospholipid membranes, suggesting that it may penetrate membranes (Hu et al., 2008). These peptides could then stimulate immunologically silent disposal of damaged membrane products.

## Materials and methods

### Cell lines, media, general chemicals, DNA, and RNA

HEK293T, C2C12 HCT116, and all HCT116-derived cell lines were grown at 37°C in 5% $CO_2$ in DMEM supplemented with 10% FBS (Thermo Fisher Scientific). Cells were routinely tested and found negative for mycoplasma contamination with the MycoAlert Mycoplasma Detection Kit (Lonza Biosciences). HCT116, C2C12, and HEK293T cells were provided and authenticated by the University of California (UC) Berkeley Cell Culture Facility using short tandem repeat (STR) profiling. For Figs. 1 and S1, HCT116 or HEK293T cells were incubated in EV-depleted medium produced by ultracentrifugation of DMEM supplemented with 25% FBS at 186,000 × g (40,000 RPM) for 24 h using a Type 45Ti rotor. Ultracentrifuged medium was then diluted to 10% FBS with DMEM. For SILAC experiments, HEK293T cells were grown in high glucose DMEM for SILAC (Athena Enzyme Systems), supplemented with 10% dialyzed FBS (Thermo Fisher Scientific) for 2 wk prior to exosome isolation. DMEM was supplemented with GlutaMax, L-leucine (800 μM), L-methionine (200 μM), and either L-lysine (800 μM) and L-arginine (400 μM) or L-lysine $^{13}C_6$, $^{15}N_2$ (800 μM) and L-arginine $^{13}C_6$, $^{15}N_4$ (400 μM). EV-depleted SILAC medium was prepared as described for EV depletion of normal medium. C2C12 cells were differentiated for 5 days in DMEM supplemented with 2% horse serum and 1 μM insulin. Antibodies for immunoblot were rabbit anti-vinculin (ab129002; Abcam), rabbit anti-annexin A6 (201024; Abcam), rabbit anti-annexin A1 (ab214486; Abcam), rabbit anti-annexin A2 (ab185957; Abcam), rabbit anti-calpain 1 (10538-1-AP; Proteintech), mouse anti-CD63 (556019; BD Biosciences), rat anti-CD63 (LS-C179520; LSBio), mouse anti-alix (sc-53540; Santa Cruz), mouse anti-flotillin-2

(610384; BD Biosciences), rabbit anti-CD9 (D801A; CST), mouse anti-tubulin (ab7291; Abcam), rabbit anti-streptolysin O (GTX64171; GeneTex), mouse anti-NanoLuc (MAB10026; R&D Systems), rabbit anti-HA (C29F4; CST), rabbit anti-LC3 (NB100–2220; Novus), and rabbit anti-Lamp1 (D2D11; CST). RNA for mRNA-seq was purified using the Direct-zol RNA Miniprep kit (Zymo Research).

## Lentivirus production and transduction

The pLJM1 (#91980; Addgene) backbone was used for stable expression of annexin A2 and S100A10 in HCT116 cells. Fusion protein domains were separated by (GGSG)$_2$ linkers. For low expression of annexin A2-Nluc (Figs. 2, S2, and S3), WT/Tr/Mt-annexin A2-mScarlet/mNeonGreen (Figs. 5 and S5), WT/Tr/Mt-annexin A2-HA (Fig. 5), and S100A10-mNeonGreen (Fig. 5), the pLJM1 CMV promoter was replaced with the RPL30 promoter. pLJM1 plasmids expressing mNeonGreen-tagged proteins contained puromycin resistance cassettes. For pLJM1 plasmids expressing mScarlet and Nluc-tagged proteins, the puromycin resistance cassette was replaced with a blasticidin resistance cassette. The pLIX403-puro (#41395; Addgene) backbone was used for stable, doxycycline-inducible mCherry-VPS4a(E228Q) expression.

HEK293T cells at 40% confluence within a 6-well plate were transfected with 165 ng of pMD2.G (#12259; Addgene), 1.35 µg of psPAX2 (#12260; Addgene), and 1.5 µg of a pLJM1 or pLIX403 plasmid using the TransIT-LT1 Transfection Reagent (Mirus Bio) as per the manufacturer's protocol. At 48 h after transfection, 1 ml of fresh DMEM supplemented with 10% FBS was added to each well. The lentivirus-containing medium was harvested 72 h after transfection by filtration through a 0.45-µm polyethersulfone filter (VWR Sciences). The filtered lentivirus was distributed in aliquots, snap-frozen in liquid nitrogen, and stored at –80°C. For lentiviral transductions, we infected HCT116 cells with filtered lentivirus in the presence of 8 µg/ml polybrene for 24 h, and the medium was replaced. HCT116 cells were selected using 1 µg/ml puromycin or 4 µg/ml blasticidin S for 4 days and 6 days, respectively. Gene expression was assessed by immunoblot analysis.

## Transfections

The pEGFP-N1 (Clontech) backbone with EGFP replaced with mNeonGreen or mScarlet was used for transient expression of annexin fusion proteins. Fusion protein domains were interleaved by (GGSG)$_2$ linkers. 1.25 µg of annexin-expressing plasmid was mixed with 125 µl of Opti-MEM (Thermo Fisher Scientific) and 2.5 µl of Lipofectamine 2000 (Thermo Fisher Scientific), vortexed briefly, and incubated for 20 min at room temperature. DNA–lipid complexes were added to cells in a 35-mm dish at 50% confluence.

For annexin A2 KO generation, a guide targeting the first annexin A2 intron (5′-CTGATACGGGATGTTGACAG-3′) was cloned into a pX330 plasmid (#98750; Addgene). A homology-directed repair template was also constructed. Moving from the 5′ to 3′ end, this construct consisted of 500 bp of the gDNA sequence N-terminal to the guide cut site, a β-globin intron splice acceptor, mTagBFP, T2A, blasticidin S resistance, SV40

poly(A), and 500 bp of the gDNA sequence C-terminal to the guide cut site. 1:1 M ratio of pX330 and the homology-directed repair template were transfected as described above. After blasticidin S selection, KO clones were generated by plating single cells into wells of 96-well plates.

## Immunoblotting

Cells were washed once with PBS and lysed in TBS containing 1% TX-100, 1 mM EGTA, and a protease inhibitor cocktail (1 mM 4-aminobenzamidine dihydrochloride, 1 µg/ml antipain dihydrochloride, 1 µg/ml aprotinin, 1 µg/ml leupeptin, 1 µg/ml chymostatin, 1 mM phenylmethylsulfonyl fluoride, 50 µM N-tosyl-L-phenylalanine chloromethyl ketone, and 1 µg/ml pepstatin) and incubated on ice for 10 min. The whole cell lysate was centrifuged at 1,000 × $g$ for 10 min at 4°C and the PNS was diluted with 6× Laemmli buffer (without DTT) to a 1× final concentration. Samples were heated at 95°C for 5 min, and proteins resolved on 4–20% acrylamide Tris-glycine gradient gels (Life Technologies). Proteins were then transferred to polyvinylidene difluoride membranes (EMD Millipore), blocked with 5% BSA in tris buffered saline with 0.1% tween-20 (TBS-T), and incubated overnight with primary antibodies in 5% BSA in TBS-T. The membranes were then washed again three times with TBS-T, incubated for 1 h at room temperature with 1:10,000 dilutions of anti-rabbit (NA934; Cytiva), anti-mouse (NXA931; Cytiva), or anti-rat (31470; Invitrogen) secondary antibodies, washed three times with TBS-T, and then detected with ECL-2 or PicoPLUS reagents (Thermo Fisher Scientific) for proteins from cell lysates or EV isolations, respectively. Immunoblots were imaged using a ChemiDoc Imaging System (Bio-Rad).

## Synchronized SLO treatment

Lyophilized SLO (Sigma-Aldrich) was pre-activated in PBS containing 10 mM DTT at 37°C for 1 h, distributed in aliquots into low-retention microcentrifuge tubes, snap-frozen in liquid nitrogen, and stored at –80°C until use. The protein concentration of the SLO batch was determined by a Bradford assay. HCT116 cells plated 2 days before (70–80% confluent) were washed with cold PBS and incubated in prechilled DMEM (w/o calcium), 1 mM EGTA, and the indicated SLO concentration for 15 min at 4°C. Cells were washed once with cold PBS and replaced with prewarmed 37°C DMEM (w/o calcium) + 1.8 mM CaCl$_2$ and incubated for 20 min at 37°C.

## Crude, high-speed EV pellet centrifugation

Conditioned medium from 8 × 15-cm plates (240 ml) was harvested from untreated (Fig. 1) or 200 ng/ml SLO-treated (Fig. S2 G; Fig. 3, A and B; and Fig. 4 E) HCT116 cells. All subsequent manipulations were completed at 4°C. Cells and large debris were removed by low-speed sedimentation at 1,000 × $g$ for 15 min in a Sorvall R6 + centrifuge (Thermo Fisher Scientific) followed by medium-speed sedimentation at 10,000 × $g$ for 15 min using a fixed angle FIBERlite F14–6×500 y rotor (Thermo Fisher Scientific). The supernatant fraction was then centrifuged at 29,500 RPM (~100k × $g$) for 1.25 h in a SW32 rotor. The high-speed pellet fractions were resuspended in PBS unless otherwise stated and pooled (600 µl total).

## EV fractionation by buoyant iodixanol density gradient equilibration

Fresh aliquots of 5.4%, 10.8%, 16.2%, and 21.6% (vol/vol) iodixanol solutions were prepared by mixing appropriate volumes of PBS and Solution D (PBS and 54% [wt/vol] iodixanol). A 27% (vol/vol) iodixanol solution was prepared by mixing the resuspended high-speed pellet fraction with Solution D. Iodixanol gradients were prepared by sequential 1 ml overlays of each iodixanol solution in a 5-ml SW55 tube, starting with the 27% iodixanol solution and finishing with the 5.4% iodixanol solution. Gradients were centrifuged in a SW55 rotor at 36,500 RPM for 16 h with acceleration set at a minimum level and no brake. Fractions (350 μl) were collected from top to bottom. Density measurements were taken using a refractometer. Each fraction was diluted in 6× Laemmli buffer (without DTT) for immunoblot analysis.

## EV fractionation by buoyant sucrose step gradient equilibration

Fresh aliquots of 10%, 40%, and 60% (vol/vol) sucrose solutions were prepared in 20 mM Tris (pH 7.4). A 55% (vol/vol) sucrose solution was prepared by mixing the resuspended high-speed pellet with the 60% sucrose solution. Sucrose gradients were prepared by sequentially layering 1 ml of the 40% and 10% sucrose solution over 3 ml of the 55% sucrose solution in a 5-ml SW55 tube. Gradients were centrifuged in a SW55 rotor at 36,500 RPM for 16 h with minimum acceleration and no brake. Fractions (400 μl) were collected from top to bottom. Density measurements were taken using a refractometer. Each fraction was diluted in 6× Laemmli buffer (without DTT) for immunoblot analysis.

For SILAC experiments, after flotation, the third and fourth fractions from the top of gradient were diluted with 10 ml PBS and centrifuged to sediment EVs. Pellet fractions were resuspended in TBS + 2% SDS, and protein concentration was determined using a microBCA kit (Thermo Fisher Scientific). Fraction 3 protein (1 μg) from heavy cells was mixed with Fraction 4 protein (1 μg) from light cells and vice versa. Glycerol (10%) was added to both mixes, and samples were loaded onto an SDS-PAGE gel. Proteins were electrophoresed into the stacking gel, and the gel was stained using Sypro Ruby using the overnight protocol (Thermo Fisher Scientific). Protein samples were excised from the gel, digested in-gel with trypsin, and analyzed by quantitative mass spectrometry. Proteomics data were analyzed using MaxQuant software.

## Exosome pulldowns

For CD63+ EV immunoprecipitation, Protein G Magnetic Dynabeads (50 μl) (Thermo Fisher Scientific) were washed twice with 200 μl PBS using a magnetic rack to capture beads between washes. The resuspended high-speed pellet fraction and 1 μg anti-CD63 (clone H5C6; BD Biosciences), were added to the beads and incubated with rotation overnight at 4°C. The beads were washed three times with 600 μl cold PBS, incubating for 1 min between washes. EVs were eluted with 30 μl PBS + 0.1% TX-100 for 5 min. Fractions were diluted in 6× Laemmli buffer (without DTT) for immunoblot analysis.

For pulldown of EVs with exposed phospholipids using annexin A5, the high-speed EV pellet fraction from 2 × 15-cm plates of SLO-treated cells was resuspended in 200 μl of annexin-binding buffer (10 mM HEPES, pH 7.4, 140 mM NaCl, and 2 mM calcium chloride). Biotin-X ANXA5 (5 μl) (Thermo Fisher Scientific) was added, mixed, and incubated for 15 min at room temperature. Annexin-binding buffer (350 μl) and 50 μl of Steptavidin Dynabeads (Thermo Fisher Scientific), prewashed with 50 μl annexin-binding buffer, were added to the binding reaction and incubated for 10 min. The beads were washed twice with 400 μl annexin-binding buffer and eluted with 200 μl of 10 mM HEPES, pH 7.4, 140 mM NaCl, 2 mM EGTA, and 0.1% TX-100. Fractions were diluted in 6× Laemmli buffer (without DTT) for immunoblot analysis.

## ANXA2-Nluc and CD63-Nluc secretion assay

Assays were performed as described previously (Williams et al., 2023), with slight modifications. HCT116 CD63-Nluc or ANXA2-Nluc cells were grown to ~80% confluence in 24-well plates. All subsequent manipulations were performed at 4°C. Conditioned medium (200 μl) was taken from the appropriate wells, added to a microcentrifuge tube, and centrifuged at 1,000 × g for 15 min in an Eppendorf 5430 R centrifuge (Eppendorf) to remove intact cells. Supernatant fractions (150 μl) from the low-speed sedimentation were passed through a 0.45-μm filter (96-well format) by centrifuging at 1,000 × g for 5 min. Filtered fractions (50 μl) were used to measure luminescence. During these centrifugation steps, the cells were placed on ice, washed once with cold PBS, and lysed in 200 μl of PBS containing 1% TX-100 and protease inhibitor cocktail.

To measure vesicular Nluc secretion, we prepared a master mix containing the membrane-permeable Nluc substrate and a membrane-impermeable Nluc inhibitor using a 1:1,000 dilution of Extracellular NanoLuc Inhibitor and a 1:333 dilution of NanoBRET Nano-Glo Substrate into PBS (Promega). Aliquots of the Nluc substrate/inhibitor master mix (100 μl) were added to 50 μl of the supernatant fraction from the medium-speed centrifugation and vortexed briefly, and luminescence was measured using a Promega GlowMax 20/20 Luminometer (Promega). For the intracellular normalization measurement, the luminescence of 50 μl of cell lysate was measured using the Nano-Glo Luciferase Assay kit (Promega) as per the manufacturer's protocol. The EV production index for each sample is calculated as follows: EV production index = medium/cell lysate.

## Isolation of cytosol from cultured human cells

Isolations were performed as described previously (Williams et al., 2023), with slight modifications. HCT116 WT cells were grown to ~90% confluence in 150-mm dishes. All subsequent manipulations were performed at 4°C. Each 150-mm dish was washed once with 10 ml of cold PBS and then harvested by scraping into 5 ml of cold PBS + 1 mM EGTA. The 5-ml cell suspensions from either 4 (Fig. 4, A and B) or 20 (Fig. 4, I and J) 150-mm dishes were combined. Cells were then collected by centrifugation at 200 × g for 5 min, and the supernatant fraction was discarded. The cell pellet was resuspended in 2 vol of lysis buffer (TBS, 1 mM EGTA, 0.2 mM PMSF, and for Fig. 4, I and J,

10 μM E64) and placed on ice. Cells were mechanically lysed by 15 strokes through a 22-gauge needle. Cell lysates were centrifuged at 1,000 × g for 15 min to sediment intact cells and nuclei, and the PNS was then centrifuged at 49,000 RPM for 15 min in a TLA-55 ultracentrifuge (Beckman Coulter). The supernatant (cytosol fraction) was collected conservatively without disturbing the pellet.

## Protein purification of ANXA2, ANXA2-S100A10 complex, and ANXA2-HaloTag

Pet28a vectors expressing ANXA2, S100A10, or ANXA2-HaloTag were transformed into Rosetta (DE3) BL21 *Escherichia coli* and grown in 250 ml LB cultures at 37°C. At O.D. 600 = 0.5, protein expression was induced with 50 μM IPTG. Cultures were grown overnight at 18°C and centrifuged at 3k × g for 10 min. Pellet fractions were kept at –80°C until time for purification. Pellet samples were thawed and resuspended in 7.5 ml *E. coli* Lysis Buffer (1x TBS, 1x protease inhibitor cocktail, 10 μl benzonase [NEB], 1 mg/ml lysozyme, 5 mM DTT, and 1 mM EGTA). Bacteria were lysed by sonication for 5 s on (20% power), 20 s off, 5 times. *E. coli* lysate was centrifuged for 5 min at 1,000 × g at 4°C. The supernatant fraction was removed and transferred to a high-speed, 1.5-ml centrifuge tube (Beckman Coulter). For purification of ANXA2–S100A10 complex only, ANXA2 lysate and S100A10 lysate were mixed 1:1. 2 mM CaCl$_2$ (final) was added and incubated for 5 min at room temperature. Lysates were centrifuged at >100k × g (49k RPM) in a TLA 55 for 10 min, and the supernatant was discarded. The membrane pellet was resuspended with a 5 ml annexin wash buffer (1x TBS, 5 mM CaCl$_2$, 5 mM ATP, and 5 mM DTT), using a 25-gauge needle for thorough resuspension. The resuspended pellet was centrifuged again at 100k × g (49k RPM) in a TLA 55 rotor for 10 min. The supernatant fraction was carefully removed, and the membrane pellet resuspended with 1 ml of annexin elution buffer (1x TBS, 5 mM DTT, and 10 mM ATP). DTT was replaced with TCEP for downstream labeling of the eluted annexin A2 with quencher. The resuspended pellet fraction was centrifuged again at 100k × g (49k RPM) in a TLA 55 for 10 min. The supernatant fraction was carefully removed, and aliquots were flash frozen in liquid nitrogen for storage at –80°C. For labeling, annexin A2-Halo (14 μM) elution was mixed with 16 μM JF646 Halo Ligand (Promega). For self-quenched annexin A2-Halo, 286 μM Tide Quencher 5WS-Maleamide (AAT Bioquest) was also added to this reaction and incubated overnight at 4°C. The maleamide reaction was quenched with 5 mM DTT, and unreacted Halo Ligand and Quencher were removed using two Bio-Spin 6 (Tris) columns (Bio-Rad). For ANXA2-S100A10 and ANXA2 only, gel filtration (Superdex-200; GE Healthcare) with TBS as gel filtration buffer was used to further purify complexes.

## Protein purification of dCAPN1 and dCAPN2

Human dCAPN1[C115S]-3xFlag-6xHis and dCAPN2[C105S]-3xFlag-6xHis were both cloned into PetDuet-1 coexpressing CAPNS1(86–269). *E. coli* cultures were grown at 37°C in 200 ml cultures to an O.D.600 of 0.5, and expression was induced with 50 μM IPTG. Cultures were grown overnight at 18°C. *E. coli* were centrifuged at 3k × g for 10 min and resuspended in 5 ml TBS +

0.2 mM PMSF. Cells were sonicated five times (5 s on, 15 s off, and 20% power), and lysates sedimented at 10k × g for 15 min. Supernatant fractions were applied to HisPure beads (500 μl of slurry) prewashed with lysis buffer. Slurries were rotated at 4°C for 1 h and then washed three times with 5 ml 1.5x TBS + 20 mM imidazole and centrifuged at 600 × g after washes to collect beads. Protein was eluted with 1 ml 1x TBS + 300 mM imidazole. Aliquots (50 μl) were snap frozen in liquid nitrogen.

## Capture of CAPN1 and CAPN2 substrates and interactors

CaCl$_2$ (2 mM final) was added to E64-containing cytosol isolated as described above and incubated for ~1 h. In parallel, 125 μl anti-Flag agarose beads (Chromotek ffa-10) were sedimented at 1k × g for 2 min and resuspended in 300 μl wash buffer (TBS + 1 mM CaCl$_2$ + 0.01% Tween-20). The resuspended beads were split into three tubes. Flag (3x) peptide, dCAPN1-3xFlag, or dCAPN2-3xFlag bait was added to the beads (6 μM final for each bait) in a 300 μl reaction. Bait proteins were bound to beads for 1 h at 4°C. Beads were washed once with 500 μl wash buffer and once with 500 μl lysis buffer (TBS + 10 μM E64 + 0.2 mM PMSF + 1 mM EGTA), with centrifugation at 1k × g for 2 min between washes to sediment beads. CaCl$_2$-containing lysate (1.5 ml) was added to each of the three conditions and incubated for 2 h at 4°C. Beads were washed three times with 1 ml wash buffer, with centrifugation at 1k × g for 2 min between washes to sediment beads. Calpain substrates were eluted with 100 μl elution 1 buffer (TBS + 5 mM EGTA), and stable calpain-interacting proteins were eluted with 100 μl elution 2 (TBS + 1x Laemmli Buffer).

## Measurement of plasma membrane permeabilization by SLO over time

For measurement of plasma membrane permeabilization in bulk, we resuspended dissociated HCT116 cells in PBS + 5 mM EGTA. Cell slurries (50 μl, 4 × 10$^5$ cells per reaction) were incubated on ice in a quantitative PCR (qPCR) plate with the indicated concentration of SLO. Cells were sedimented at 300 × g and resuspended in 50 μl of ice-cold PBS + 5 mM EGTA + 2.5 μM SYTOX Green (S7020; Thermo Fisher Scientific). The plate was sealed with optical film and placed in a qPCR machine (CFX96; Bio-Rad) pre-cooled to 4°C and then rapidly heated to 37°C with fluorescence measured every 20 s using SYBR green settings. For measurement of plasma membrane permeabilization with microscopy, we grew HCT116 cells to 70% confluence in a 35-mm glass bottom dish (MatTek). Cells were placed in a 37°C, 5% CO$_2$ chamber on an LSM900 microscope and then washed once with 1 ml Ca$^{2+}$-free DMEM and incubated with Ca$^{2+}$-free DMEM + 200 ng/ml SLO + 2.5 μM SYTOX Green. Where indicated, 1 mM Ca$^{2+}$ was added to the incubation media. SYTOX Green uptake was measured by imaging with the 10× objective, taking images every 10 s for 10 min.

## GUVs and liposomes

For giant unilemellar vesicle (GUV) assays, we mixed lipids in the following molar ratios: 38.5:20:20:3:18.5 DOPC:POPE:DOPS:PI(4,5)P$_2$:cholesterol (1 μmol total lipid) in 500 μl of 5% methanol and 95% chloroform. The lipid mix was smeared onto two indium tin oxide-coated glass plates, and for 30 min, solvent was

evaporated under vacuum on a heat block preheated to 55°C. A circular rubber spacer coated with vacuum grease was placed on the glass slide into which buffer was introduced (1 mM Tris, pH 7.4, and 250 mM sucrose), followed by covering with a second glass slide. Electrodes were clipped to each plate, and the chamber was placed in a 50°C oven. A function generator was used to apply an electric field (10 Hz, 1.4 V) for 90 min. For the last 30 min, the frequency was decreased by 0.5 Hz every 5 min. The GUV-containing solution was drained from the chamber and mixed with GUV buffer (5 mM Tris, pH 7.4 and 250 mM glucose). After settling to the bottom of the slide overnight at 4°C, GUVs were resuspended in GUV buffer with 2 mM $CaCl_2$ and 5 mM TCEP and mixed with combinations of 1 µM ANXA2-Halo-JF646 and 1 µM S100A10-mScarlet.

For liposome assays, we mixed lipids in a glass beaker in the following molar ratios: 38:20:20:3:0.5:18.5 DOPC:POPE:DOPS:PIP(4,5)$P_2$:TexasRed-PE:cholesterol (1 µmol total lipid) in 500 µl of 5% methanol and 95% chloroform. The lipid mix was placed on a heat block preheated to 55°C, and solvent was evaporated under vacuum for 30 min. Lipids were resuspended in degassed TBS and incubated at 55°C with intermittent vortexing for 15 min. The lipid solution was pumped 15 times through an extruder with a 200-µm filter. Liposomes were diluted 1:5 into TBS with 2 mM $CaCl_2$ and 5 mM TCEP, with indicated combinations of 300 nM untagged ANXA2, ANXA2–S100A10 complex, and porcine CAPN-I.

### Microscopy and laser ablation

The images in Fig. 5, I and J; Fig. S2 E; and Fig. S5 D were acquired on an Echo Revolve Microscope using the 10× air objective. For image quantification in Fig. 5, I and J, the "Analyze Particles" function in ImageJ was used to quantify aggregates area. The images in Fig. 2, A and C; Fig. S2, A–D; Fig. 4 H; Fig. 5, C, E, and G; Fig. S5 B; and Fig. 6 D were acquired using an LSM900 confocal microscope system (ZEISS) using confocal mode, a 63× Plan-Apochromat, NA 1.40 objective, and a heated (37°C), $CO_2$-controlled chamber. Zen 3.1 (Zeiss) software was used for acquisition. Cells were bathed in DMEM with 2.5 µM FM1-43 (Biotium) added where indicated. For laser ablations using the LSM900, a 1 × 1 µM square was positioned over the edge of a cell not adjacent to another cell. The 1 × 1 µM square was ablated for 100–200 iterations using the UV laser at 100% power.

Repair caps were quantified in Zen 3.1 (Zeiss) by measuring intensities within a box that encapsulated the repair cap. This intensity was normalized to a boxed piece of membrane that was not ablated.

### Online supplemental material

Fig. S1 shows that the annexin-containing EVs are distinct from exosomes. Fig. S2 shows that the annexin-containing EVs are shed from the repair scab after plasma membrane damage. Fig. S3 shows that the annexins within EVs are shifted in apparent molecular weight. Fig. S4 shows that the calpain-1/2 cleaves annexins, which are then shed in MVs. Fig. S5 shows that the calpain cleavage attenuates the membrane binding and scabbing activity of annexin A2. Table S1 provides the ratios and P values for the enrichment of EV proteins in the low vs. high buoyant density fractions of a sucrose gradient. Table S2 lists the peptides detected from gel-excised annexin A2 protein treated with (Tab 2) or without (Tab 1) porcine calpain-1. Table S3 provides the protein abundances, calculated by peptide count, normalized spectral abundance factor, or exponentially modified protein abundance index for EGTA elutions from pulldown reactions using 3xFlag, 3x-Flag C115S calpain-1 (CAPN1), or 3x-Flag C105S calpain-2 (CAPN2) as bait. Video 1 shows the time-lapse confocal microscopy of shedding from the repair scab of laser-ablated (100 iterations), FM1-43–stained (2.5 µM, gold) HCT116 cells. Video 2 shows the time-lapse confocal microscopy of shedding from the repair scab of laser-ablated (200 iterations), FM1-43–stained (2.5 µM, gold) HCT116 cells.

### Data availability

All proteomics datasets are available in the supplementary material. Raw proteomics and mRNA-seq data generated during the current study are available from the corresponding author upon reasonable request.

## Acknowledgments

We dedicate this work to Bob Lesch, our lab manager for the past several decades, who was tragically taken from us by an accident in 2021. We would also like to thank our current lab manager, Nam Che, and the staff at the UC Berkeley shared facilities, the Cell Culture Facility (Alison Killilea) and the DNA Sequencing Facility. The mCherry-VPS4a (dominant mutant) cDNA was a gift from Kevin Rose in the laboratory of James H. Hurley (UC Berkeley, Berkeley, CA, USA).

This work used the Vincent J. Proteomics/Mass Spectrometry Laboratory at UC Berkeley, RRID:SCR_025852, and QB3 Genomics, UC Berkeley, Berkeley, CA, USA, RRID:SCR_022170. J.M. Ngo is supported by a National Institutes of Health F31 grant. R. Schekman is an investigator of the Howard Hughes Medical Institute, a senior fellow of the UC Berkeley Miller Institute of Science, and chair of the Scietnific Advisory Board of Aligning Science Across Parkinson's Disease. This work was funded by the Howard Hughes Medical Institute. The funders had no role in study design, data collection and interpretation, or the decision to submit the work for publication.

Author contributions: J.K. Williams: conceptualization, data curation, formal analysis, investigation, methodology, software, supervision, validation, visualization, and writing—original draft, review, and editing. J.M. Ngo: conceptualization, formal analysis, investigation, and methodology. A. Murugupandiyan: investigation. D.E. Croall: methodology. H.C. Hartzell: conceptualization, investigation, methodology, validation, and writing—review and editing. R. Schekman: conceptualization, funding acquisition, project administration, supervision, and writing—review and editing.

Disclosures: The authors declare no competing interests exist.

Submitted: 22 August 2024

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

# Supplemental material

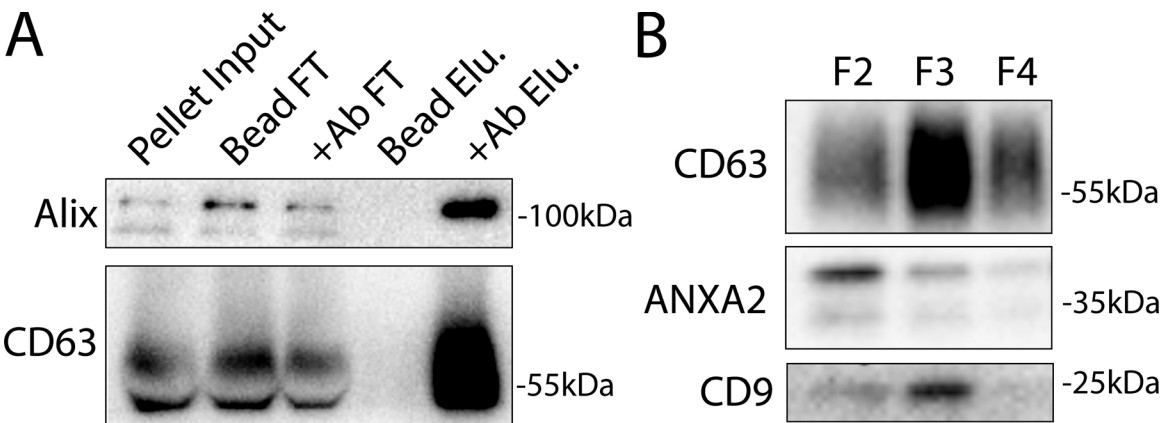

Figure S1. **Annexin-containing EVs are distinct from exosomes. (A)** Immunoblots show enrichment of exosome proteins from the conditioned medium 100k × *g* pellet (pellet input) after immunoprecipitation with protein G beads when anti-CD63 antibody (Ab) is added to the binding reaction (FT—binding reaction flow through; Elu—binding reaction elution). **(B)** Immunoblots show the distribution of EV markers across a sucrose step gradient of the conditioned medium 100k × *g* pellet fraction. Samples were taken from low density (F2-Fraction #2) to high density (F4-Fraction #4). Source data are available for this figure: SourceData FS1.

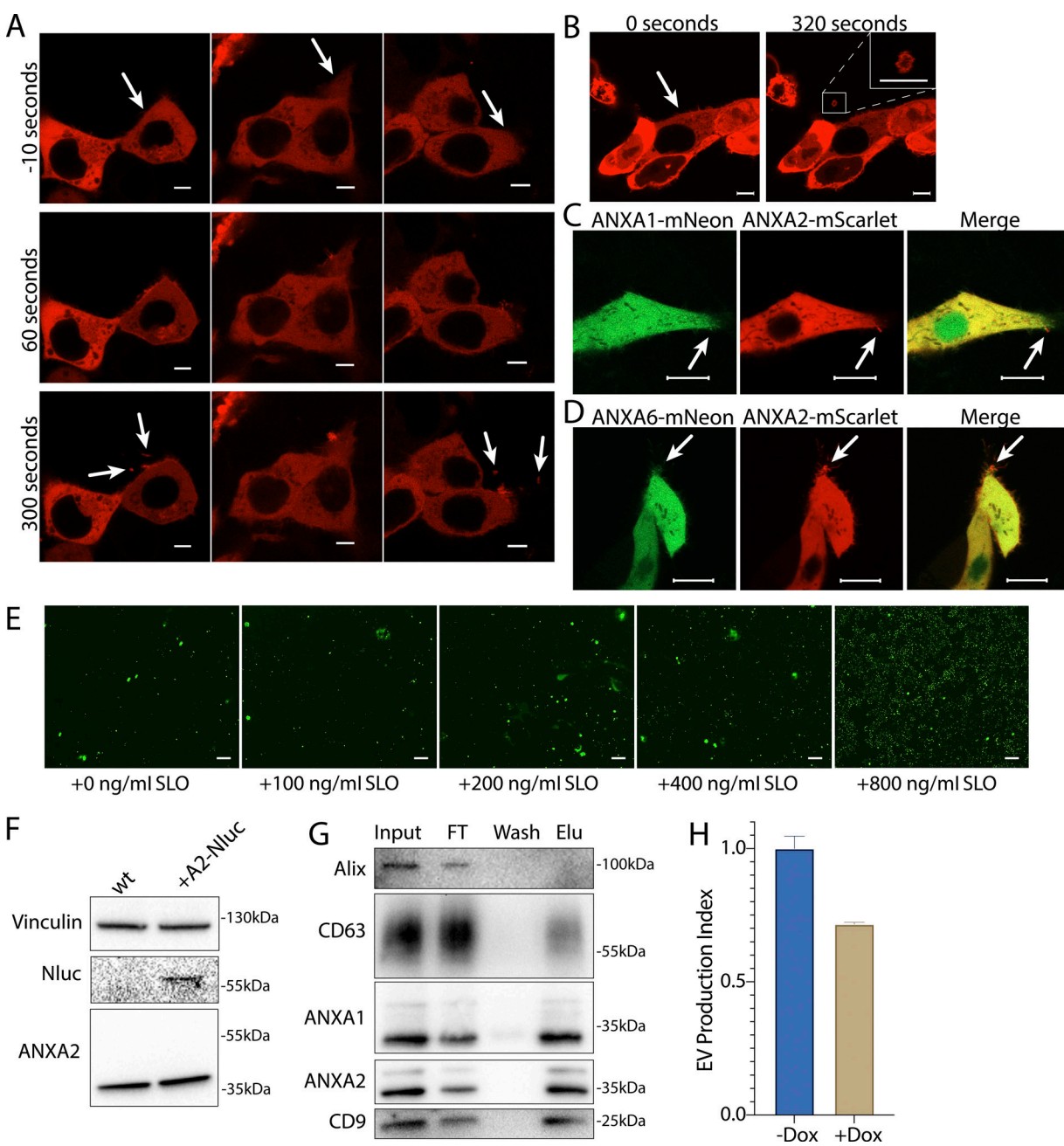

Figure S2. **Annexin-containing EVs are shed from the repair scab after plasma membrane damage. (A)** Confocal micrographs of ANXA2-mScarlet recruitment in three laser ablation experiments are shown. Image times are relative to the first image taken after ablation. White arrows in the top panes indicate the sites of ablation. Arrows in the bottom panes indicate EVs. Scale bars: 5 µm. **(B)** Representative confocal micrographs of ANXA2-mScarlet shedding are shown. Image times are relative to the first image taken after ablation. White arrows in panel indicate the site of ablation. Scale bars: 5 µm. **(C)** Representative confocal micrographs of ANXA2-mScarlet and ANXA1-mNeonGreen–expressing cells after laser ablation. Arrows indicate the ablation site. Scale bars: 10 µm. **(D)** Representative confocal micrographs of ANXA2-mScarlet and ANXA6-mNeonGreen–expressing cells after laser ablation. Arrows indicate the ablation site. Scale bars: 10 µm. **(E)** Representative widefield micrographs of cells stained with 1 µM Sytox Green after a treatment period with the indicated SLO concentration and recovery period. Scale bars: 150 µm. **(F)** Immunoblots show expression of annexin A2-Nluc (A2-Nluc) using a low expression promoter. **(G)** Immunoblots show enrichment of EV markers after capture with immobilized annexin A5 from the conditioned medium 100k × g pellet fraction (FT—flow through; Elu—elution). **(H)** EV production index from ANXA2-Nluc cells expressing mCherry-VPS4a (dominant mutant) under control of a doxycycline-inducible promoter. Cells were pretreated with 200 ng/ml doxycycline (Dox) or DMSO for 6 h, followed by treatment with 200 ng/ml SLO. Error bars indicate three experimental replicates. Source data are available for this figure: SourceData FS2.

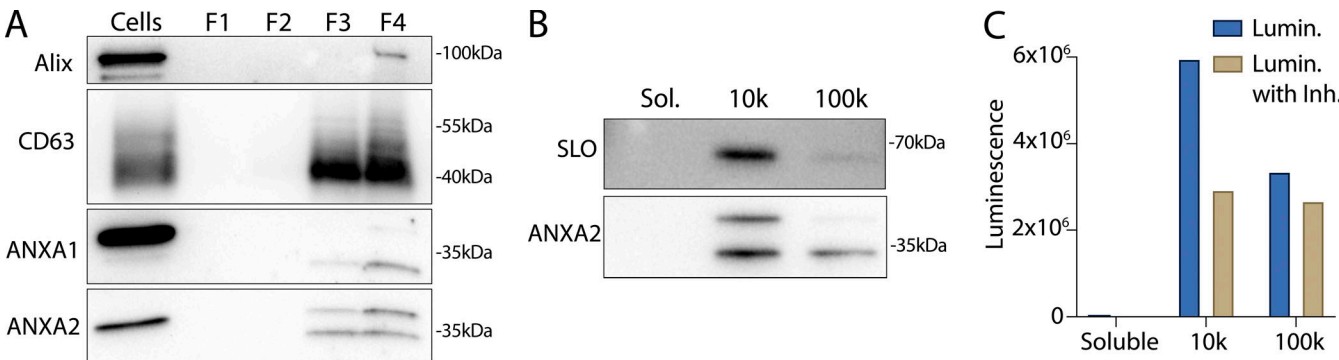

Figure S3. **Annexins within EVs are shifted in apparent molecular weight. (A)** Immunoblots show enrichment of indicated EV markers relative to cell lysate after treatment of C2C12 myotubes with 200 ng/ml SLO. F1, F2, F3, and F4 refer to buoyant fractions of a sucrose step gradient of the conditioned medium 100k × *g* pellet fraction, moving from low to high density. **(B and C)** (B) Immunoblot and (C) luminescence analysis of the 10k × *g* pellet fraction (10k), the 100k × *g* pellet fraction (100k), and the remaining soluble supernatant (Sol.) after serial centrifugation of conditioned media from ANXA2-Nluc cells treated with 200 ng/ml SLO. For each fraction, Nluc luminescence (lumin.) was measured with or without membrane-impermeable Nluc inhibitor (Inh.). Source data are available for this figure: SourceData FS3.

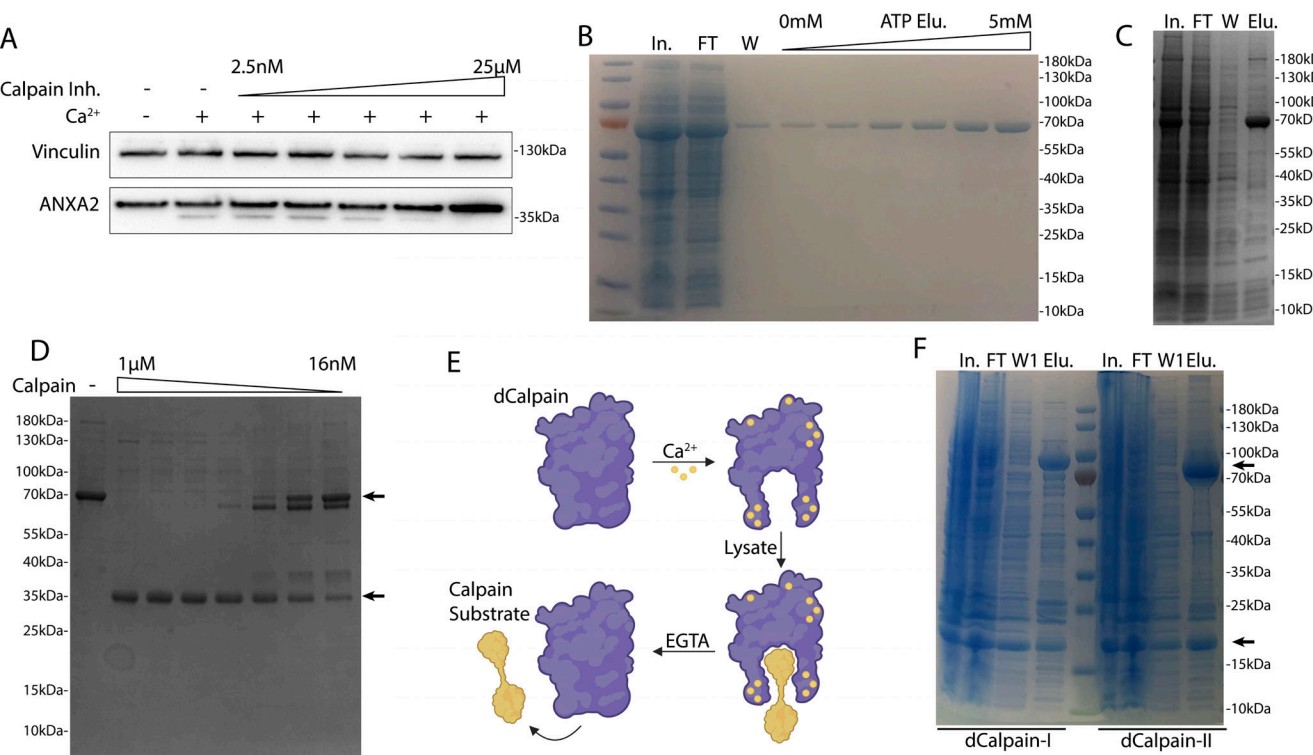

Figure S4. **Calpain-1/2 cleave annexins, which are then shed in MVs. (A)** Immunoblot analysis of cytosol fractions after incubation with or without 1 mM Ca²⁺, with a range of concentrations of ALLN inhibitor (Calpain Inh.). **(B)** Coomassie-stained gel showing purification of annexin A2-Halo from *E. coli* (In—lysate input; FT—100k × *g* pellet fraction flow through; W—CaCl₂-containing 100k × *g* pellet wash; ATP Elu—elution from 100k × *g* pellet fraction with increasing concentrations of ATP). **(C)** Coomassie-stained gel showing purification of annexin A6-HA from *E. coli* (In—lysate input; FT—100k × *g* pellet flow through; W—CaCl₂-containing 100k × *g* pellet wash; Elu—elution off 100k × *g* pellet with 10 mM ATP). **(D)** Coomassie-stained gel showing the mobility of recombinant annexin A6-HA, incubated with a range of concentrations of purified, porcine calpain-1. Arrows indicate uncleaved and cleaved products. **(E)** Schematic illustrating substrate binding and elution of substrates (yellow) to catalytic cysteine-to-serine mutant calpain baits (dCAPN, purple). **(F)** Coomassie-stained gel showing initial his-tag purification of CAPN1[C115S]-3xFlag-6xHis (dCalpain-I) and CAPN2[C105S]-3xFlag-6xHis (dCalpain-II) in complex with CAPNS1(86–268) from *E. coli* (In—lysate input; FT—bead flow through; W—bead wash; Elu—elution from Ni²⁺ with 300 mM imidazole). Arrows indicate calpain proteins. Source data are available for this figure: SourceData FS4.

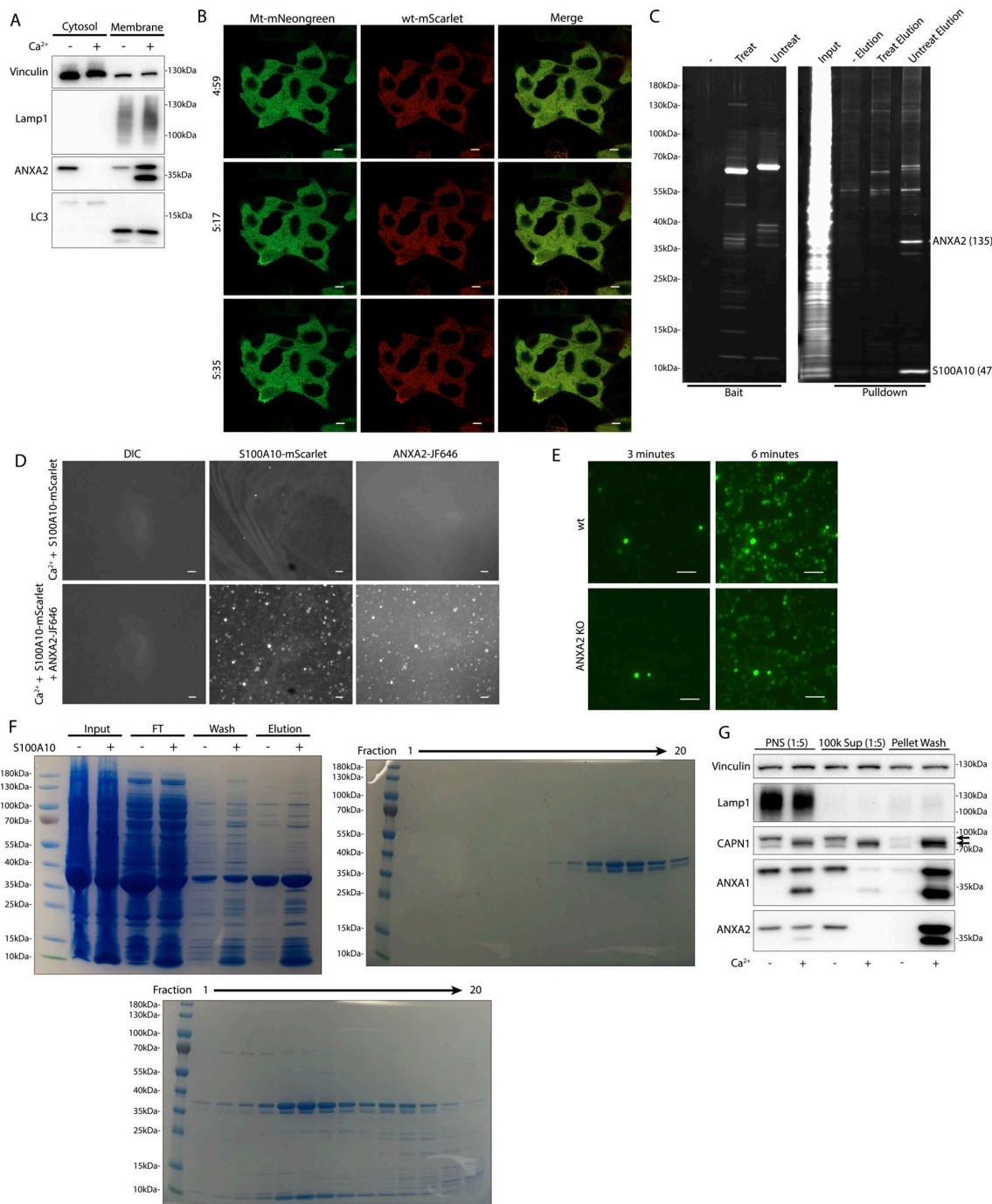

Figure S5. **Calpain cleavage attenuates the membrane binding and scabbing activity of annexin A2. (A)** Immunoblot analysis of cytosol and membrane fractions, with or without 1 mM Ca$^{2+}$ added prior to fractionation. **(B)** Representative confocal micrographs of ANXA2-mScarlet (wt-mScarlet) and ANXA2 [P20D, P21D]-mNeonGreen (Mt-mScarlet)–expressing cells. Image times are relative to the addition of 400 ng/ml SLO. Scale bars: 5 µm. **(C)** Total protein (Sypro Ruby staining) analysis of HaloTag "pulldown" on chloroalkane-coated beads is shown, using no bait (–), porcine calpain-1–treated annexin A2-HaloTag bait (treat), or untreated annexin A2-HaloTag bait (untreat). Proteins are labeled with spectral counts, in parentheses, from gel excision-mass spectrometry. **(D)** Representative widefield micrographs of purified S100A10-mScarlet (1 µM) binding to GUVs with or without annexin A2-HaloTag-JF646 (1 µM). Scale bars: 50 µm. **(E)** Representative widefield micrographs of WT or annexin A2 KO cells treated with 200 ng/ml SLO and 2.5 µM Sytox Green without Ca$^{2+}$ in the media. Scale bars: 100 µm. **(F)** Coomassie-stained gels showing purification of annexin A2 or annexin A2–S100A10 complex from *E. coli*. Panel I shows the initial purification by eluting annexin A2 from the *E. coli* lysate pellet fraction in a Ca$^{2+}$-dependent manner (Input—lysate input; FT—100k × g pellet flow through; Wash—CaCl$_2$-containing 100k × g pellet wash; Elution—10 mM ATP elution from 100k × g pellet fraction). Where indicated lysate from S100A10-expressing *E. coli* was mixed 1:1 with annexin A2 *E. coli* lysate before purification. Panels II and III show size exclusion chromatography fractions of annexin A2 and annexin A2–S100A10 complex, respectively. **(G)** Immunoblot analysis shows mechanically lysed cells sequentially fractionated into PNS (diluted 1:5 before loading), 100k × g centrifuged PNS supernatant (100k Sup, diluted 1:5 before loading), and 100k × g pellet washed with 5 mM EGTA and recentrifuged (Pellet Wash). Samples where 1 mM Ca$^{2+}$ was initially added to PNS are indicated. Arrows indicate uncleaved and autolyzed CAPN1. Source data are available for this figure: SourceData FS5.

Video 1.   **Time-lapse confocal microscopy of shedding from the repair scab of laser-ablated (100 iterations), FM1-43–stained (2.5 µM, gold) HCT116 cells.** Image times are relative to the first image taken after ablation. Arrow indicates the site of ablation. Total imaging time: 8 min. Time between acquisitions: 10 s. Video frame rates: 5 frames per second (fps). Scale bars: 5 µm.

Video 2.   **Time-lapse confocal microscopy of shedding from the repair scab of laser-ablated (200 iterations), FM1-43–stained (2.5 µM, gold) HCT116 cells.** Image times are relative to the first image taken after ablation. Arrow indicates the site of ablation. Total imaging time: 8 min. Time between acquisitions: 10 s. Video frame rates: 5 fps. Scale bars: 5 µm.

**Provided online are Table S1, Table S2, and Table S3. Table S1 provides the ratios and P values for the enrichment of EV proteins in the low vs. high buoyant density fractions of a sucrose gradient. Table S2 lists the peptides detected from gel-excised annexin A2 protein treated with (Tab 2) or without (Tab 1) porcine calpain-1. Table S3 provides the protein abundances, calculated by peptide count, normalized spectral abundance factor, or exponentially modified protein abundance index for EGTA elutions from pulldown reactions using 3xFlag, 3x-Flag C115S calpain-1 (CAPN1), or 3x-Flag C105S calpain-2 (CAPN2) as bait.**

