## [Peer Review File · The Journal of Cell Biology]

Calpains Orchestrate Secretion of Annexin-containing Microvesicles during Membrane Repair

Justin Williams, Jordan Ngo, Abinayaa Murugupandiyam, Dorothy Croall, H. Criss Hartzell, and Randy Schekman

Corresponding Author(s): Randy Schekman, University of California, Berkeley

Review Timeline:

Submission Date:	2024-08-22
Editorial Decision:	2024-10-08
Revision Received:	2025-03-04
Editorial Decision:	2025-03-15
Revision Received:	2025-03-28

Monitoring Editor: William Bement

Scientific Editor: Dan Simon

Transaction Report:

DOI: <https://doi.org/10.1083/jcb.202408159>

October 8, 2024

Re: JCB manuscript #202408159

Dr. Randy Schekman
University of California, Berkeley
Department of Molecular and Cell Biology University of California at Berkeley 482 Li Ka Shing Center #3370
Berkeley, CA 94720-3202

Dear Dr. Schekman,

Thank you for submitting your manuscript entitled "Calpains Orchestrate Secretion of Annexin-containing Microvesicles during Membrane Repair." The manuscript was assessed by expert reviewers, whose comments are appended to this letter. We invite you to submit a revision if you can address the reviewers' key concerns, as outlined here.

You will see that all three reviewers found your study to be thorough and interesting but also raised several important points that should be addressed to further strengthen this work. Reviewer #1 asks about the potential impact of calpain cleavage on other annexins and we agree that this should be addressed with new experiments. Other comments ask for additional quantifications, data clarifications, and additional discussion points. These all seem straightforward to us and we hope you will be able to address all of these comments.

GENERAL GUIDELINES:

Text limits: Character count for an Article is < 40,000, not including spaces. Count includes title page, abstract, introduction, results, discussion, and acknowledgments. Count does not include materials and methods, figure legends, references, tables, or supplemental legends.

Figures: Articles may have up to 10 main text figures. Figures must be prepared according to the policies outlined in our Instructions to Authors, under Data Presentation, <https://jcb.rupress.org/site/misc/ifora.xhtml>. All figures in accepted manuscripts will be screened prior to publication.

Supplemental information: There are strict limits on the allowable amount of supplemental data. Articles may have up to 5 supplemental figures. Up to 10 supplemental videos or flash animations are allowed. A summary of all supplemental material should appear at the end of the Materials and methods section.

Please note that JCB now requires authors to submit Source Data used to generate figures containing gels and Western blots with all revised manuscripts. This Source Data consists of fully uncropped and unprocessed images for each gel/blot displayed in the main and supplemental figures. Since your paper includes cropped gel and/or blot images, please be sure to provide one Source Data file for each figure that contains gels and/or blots along with your revised manuscript files. File names for Source Data figures should be alphanumeric without any spaces or special characters (i.e., SourceDataF#, where F# refers to the associated main figure number or SourceDataFS# for those associated with Supplementary figures). The lanes of the gels/blots should be labeled as they are in the associated figure, the place where cropping was applied should be marked (with a box), and molecular weight/size standards should be labeled wherever possible. Source Data files will be made available to reviewers during evaluation of revised manuscripts and, if your paper is eventually published in JCB, the files will be directly linked to specific figures in the published article.

The typical timeframe for revisions is three to four months. If you anticipate any difficulties in meeting this aforementioned revision time limit, please contact us and we can work with you to find an appropriate time frame for resubmission. Please note that papers are generally considered through only one revision cycle, so any revised manuscript will likely be either accepted or rejected.

Thank you for this interesting contribution to Journal of Cell Biology. You can contact us at the journal office with any questions at cellbio@rockefeller.edu.

Sincerely,

William Bement, PhD
Monitoring Editor
Journal of Cell Biology

Dan Simon, PhD
Scientific Editor
Journal of Cell Biology

Reviewer #1 (Comments to the Authors (Required)):

The authors examine the role of annexin A2 in membrane repair, with focus on annexin A2 shedding into microvesicles. The work is conducted in cultured cells and identifies the accumulation of annexin A2 at sites of plasma membrane disruption, along with accumulation into microvesicles emanating from this wounded sites. The authors focus on demonstrating the recruitment of annexin A2 to disrupted plasma membrane lesions, and this has been well demonstrated before in a number of plasma membrane settings as being important for membrane repair and resealing. It is reassuring to see the replication of these findings. More novel is the suggestion that calpain cleavage is important for the disassembly of the annexin "scab" and that this cleavage affects annexin A2 sequestration into the MVs and resolving the scab.

1. The proteomic data of EVs demonstrates that other annexins are present. This study focuses primarily on annexin A2, when other annexins have been suggested to serve as the pioneer annexins when forming a scab, consistent with the notion these other annexins may form the core of the scab while A2 maybe more peripheral in the scab. In dissolving the scab, is annexin A2 felt to be more in the periphery of the scab? If so, does this more peripheral scab location make it critical for scab dissolution because removing the peripheral component then enables removing the core components? This could be assessed by looking at other annexins or even EHD4 as an even more peripheral scab component. Are annexin A2 components more likely to be shed in early vesicle released from the site, consistent with their more peripheral scab position?
2. What is the calcium sensitivity of release of annexin A2 into the EVs? How does this affect the calpain mediated cleavage? The other annexins are also cleaved. How do their cleavage sites affect scab resolution and MV formation?
3. SLO mediates very different wound sites than other methods (e.g. laser). The authors should have more caution in interpreting these data.
4. The comparison to cutaneous wound healing is not helpful since cutaneous wound repair implicates so many different cell types, including platelets and extracellular proteins. The authors are advised to not draw analogy and simply focus on plasma membrane repair in schematic in Figure 6.
5. What was the rationale for using HCT-116 cells? It would be also helpful to clarify to the readers what this cell line is.

Reviewer #2 (Comments to the Authors (Required)):

This study provides novel insights into plasma membrane repair mechanisms by demonstrating that annexin-containing microvesicles (MVs) are secreted in response to calcium influx following membrane damage. The findings highlight the role of calpains in cleaving annexins, which facilitates release of membrane scabs and the secretion of damaged membrane and annexins as MVs.

I find the study thorough and quite compelling. The analogy between organismal wound healing, where scabs are removed after healing, and the repair of single-cell membranes where annexin scabs are removed by calpains is particularly interesting and could be emphasized more strongly in the abstract.

I only have a few comments/suggestions for improvements:

I understand that the focus here is secretion of MVs during membrane repair. However, many reports show that damaged membranes are also internalized upon injury and repair by endocytic mechanisms (e.g. upon SLO treatment) and recycled in the intracellular stomach. This aspect should be included in the text, since annexin/calpain containing scabs may also be internalized and subsequently digested in lysosomes.

In Figure 5i: "Addition of annexin A2 or, to a larger extent, addition of annexin A2-S100A10 caused liposomes to aggregate (Figure 5i). Addition of calpain-1 dissolved annexin A2 and annexin A2-S100A10 liposome aggregates."
Please provide some quantification of this!

Annexins are known to induce strong membrane curvature. The combination of membrane crosslinking and curvature induction around the wound site could form a strong platform for subsequent secretory or internalization events of damaged membrane (and scab). This aspect of annexin-mediated repair should also be included in the manuscript.

Reviewer #3 (Comments to the Authors (Required)):

The authors discuss the role of calpain in the repair of wounded cell membranes. Following injury, calcium influx triggers the recruitment of annexins to the wound site, where they form complexes with membrane lipids to plug the wound. This initial patch is later released as microvesicles (MVs). Calpains cleave annexins, detaching them from the MV membrane, thus facilitating the release of MVs. The authors utilize advanced techniques to demonstrate the molecular interactions involved in this process. However, there are several areas requiring improvement, as listed below:

1. Page 5, Line 1: Previous experiments using pulse-laser ablation are conducted over extremely short durations (~10 ms). Longer laser exposure, such as 1 minute, may cause significant damage to cytoplasmic and membrane molecules. Furthermore, conventional pulse-laser ablation disrupts not only the cell membrane but also the underlying actin cortex. A recent technique utilizing a gold-coated substrate can target only the cell membrane (Cells, 13, 341, 2024).
2. Page 5, Line 3: When does the wound pore close? The closing time might be estimated by tracking the incorporation of FM dye or SYTOX Green fluorescence.
3. Page 5, Line 4: The authors state that vesicles are generated at the wound site. However, the provided video and figures are unconvincing. Please include more definitive evidence of vesicle formation from the damaged membrane.
4. Page 5, Paragraph 4: Why do SLO pores remain open?
5. Page 6, Paragraph 3: Please specify the molecular weight of each cleaved annexin in both the text and figures.
6. Page 6, Paragraph 1: Is the molecular weight of the cleaved annexins in cell lysates identical to those found in MVs? Please clarify their molecular weights.
7. Page 6, Paragraph 2: Does the calpain inhibitor prevent wound repair? Does calpain accumulate at the wound site?
8. Figure 6: Please add a timeline indicating the opening and closing of the wound pore, the accumulation of annexin, calpain, and S100A10, as well as the generation and release of MVs.
9. Figure 6F: This figure lacks clarity. Where does the membrane for the scab originate? Since annexins are located inside MVs, the membrane is unlikely to come from preexisting intracellular vesicles such as lysosomes or other organelles. If the membrane is derived from the wounded cell membrane, FM dye fluorescence would not increase at the wound site. It is plausible that new membrane vesicles are formed, as supported by recent findings in Dictyostelium cells.
10. Page 10, Paragraph 2: Calpain inhibition or the expression of uncleavable annexin significantly reduces MV secretion. However, the inhibition is modest, suggesting that additional mechanisms contribute to MV secretion.
11. Discussion: Previous studies indicate that calpains disrupt the actin cortex to promote wound repair. Please include a discussion on how the present model relates to these earlier findings.
12. Discussion: Annexins contain annexin repeat domains that bind lipids, which persist after N-terminal cleavage by calpain. Please explain why cleaved annexins are released from the MV membrane.
13. Discussion: Is the release of wound scabs outside a common mechanism in eukaryotic cells? Please expand on this in the discussion.
14. Discussion: The same authors previously proposed that multivesicular bodies (MVBs) play a role in cellular wound repair (Williams et al., 2003, eLife). This is not mentioned in the current paper. Please include a discussion of the relationship between MVBs and MVs in wound repair.
15. Introduction, Line 5: Replace "intracellular" with "intercellular."
16. Introduction, Line 25: Reference number 12 should be updated to 13.
17. Page 4, Results: Correct "tritonX-100" to "Triton X-100."
18. Page 5: Correct "1 μ M" to "1 μ m."

Dear Dr. Bement and Dr. Simon,

Thank you for considering our manuscript for publication in *Journal of Cell Biology* and for providing us the opportunity to address the comments and suggestions of the reviewers. We believe that the comments from you and the reviewers were fair and constructive and rigorous. The comments led us to additional experiments and revisions, which we have specifically detailed below. We thank you for your consideration of our manuscript.

Reviewer #1:

1. The proteomic data of EVs demonstrates that other annexins are present. This study focuses primarily on annexin A2, when other annexins have been suggested to serve as the pioneer annexins when forming a scab, consistent with the notion these other annexins may form the core of the scab while A2 maybe more peripheral in the scab. In dissolving the scab, is annexin A2 felt to be more in the periphery of the scab? If so, does this more peripheral scab location make it critical for scab dissolution because removing the peripheral component then enables removing the core components? This could be assessed by looking at other annexins or even EHD4 as an even more peripheral scab component. Are annexin A2 components more likely to be shed in early vesicle released from the site, consistent with their more peripheral scab position?

The roll of other annexins is very interesting to us. To address this question, we performed an mRNA-seq and live cell imaging experiments (discussed below). We have included discussion of these experiments in the main text.

We could not find evidence in the literature that annexins have significant localization differences within the repair scab. Whereas differences in the recruitment timing have been reported, the exact order of recruitment is inconsistent between studies. Annexin A2 is most often the first annexin recruited to the repair scab, although some studies suggest it is the last annexin (extensively reviewed: Kayejo et al. *CTD* 2023). In myotubes, annexin A2 is one of the first annexins that localizes to the repair cap core, arriving before annexin A1 and A6 (Demonbreun et al. *JCB* 2016, Foltz et al. *JCB* 2021). Between annexin A1, A2, A4, and A6, annexin A2 translocation to the plasma membrane is the most sensitive to calcium elevation (Monastyrskaya, et al. *Cell Calcium* 2007). When cells express S100A10, annexin A2 is even more sensitive to calcium influx. We suspect that annexin A2 is the most abundant annexin within these repair structures of our cells for the following reasons:

- A. Overall cellular abundance: To directly access annexin abundance in our HCT116 model we performed mRNA-seq on wt HCT116 cells (Figure 2b). We found ~8 times more reads mapping to annexin A2 compared to the next most abundant annexin, annexin A1. Indeed, there were twice as many annexin A2 reads as the rest of the

annexins combined. This difference is not due to a difference in transcript size, as annexin A1, A2, A3, and A5 are all ~1.5 kilobases in length. Although, there is not a perfect correlation between transcript and protein abundance, this data suggests that annexin A2 is the most abundant annexin in our cell line. The abundance of different annexins likely varies between cell lines and may change the importance of annexin A2 during membrane repair.

- B. Recruitment and localization: To visualize differences in annexin localization, we cotransfected HCT116 cells with annexin A2-mScarlet and annexin A1 or annexin A6-mNeonGreen. We saw overall less recruitment of annexin A1 to the repair site of laser ablated cells compared with annexin A2 (Figure S2c). This data is similar to observations in myotubes (Foltz et al. *JCB* 2021). We also visualized substantially less annexin A1 shedding from the repair scab compared to annexin A2. Similarly, less annexin A1 is recruited to membrane fractions in the presence of calcium (Figure S5g), and annexin A1 is far less enriched in EVs after SLO treatment (Figure S3a) compared to cellular levels. We also saw less annexin A6 recruitment to the repair scab compared to annexin A2 (Figure S2d).

2. What is the calcium sensitivity of release of annexin A2 into the EVs? How does this affect the calpain mediated cleavage? The other annexins are also cleaved. How do their cleavage sites affect scab resolution and MV formation?

Good question. Annexin A2⁺ EV release requires ~100 μM Ca^{2+} in the medium for full release. This is the maximum concentration the repair scab experiences, and the average within the scab may be lower. Calpain 1 is fully active ~10-50 μM . Thus, calpain 1 is expected to be fully active at these concentrations. Calpain 2 is fully active at ~400 μM , but acquires a calcium sensitivity similar to calpain 1 in the presence of the plasma membrane lipid, $\text{PI}(4,5)\text{P}_2$. Thus, calpain 2 is also possibly fully active at these concentrations.

For the reasons discussed in response to question #1, we believe that annexin A2 is the most abundant annexin within the repair caps of damaged HCT116 cells. In other cell lines, however, other annexins may be more abundant. For all tested annexins (A1, A2, and A6), calpain cleavage is predicted to terminate membrane-bridging activity. For the single annexin domain proteins like annexin A2, calpains cleave at the N-terminus, preventing dimerization. For annexin A6, a tandem annexin domain protein, calpain cleaves between the annexin domains (Figure 3c, S4d). Thus, no matter the annexin, calpain cleavage should terminate the membrane-bridging activity of the annexin, potentially promoting scab breakdown.

3. SLO mediates very different wound sites than other methods (e.g. laser). The authors should have more caution in interpreting these data.

We agree. We have added a discussion of this in the text and suggested caution when directly comparing the results of these two types of wounds. In the discussion, we now detail which conclusions drawn from SLO or laser wounding so that interpretations are no longer conflated.

4. The comparison to cutaneous wound healing is not helpful since cutaneous wound repair implicates so many different cell types, including platelets and extracellular proteins. The authors are advised to not draw analogy and simply focus on plasma membrane repair in schematic in Figure 6.

We understand the reviewer's concern, however other reviewers like the analogy given in Figure 6 and want it emphasized more strongly. We have altered the schematic to make it clearer. We hope that helps alleviate this reviewer's concern.

5. What was the rationale for using HCT-116 cells? It would be also helpful to clarify to the readers what this cell line is.

We previously used this cell line to show that cancer cells secrete exosomes in response to membrane damage, and we wanted to use the same system to assess damage-induced MV secretion. In general, we are interested in how cancer cells secrete vesicles, as cancer cells are reported to secrete more vesicles than non-transformed cells. Additionally, it has been reported that cancer cells also sustain membrane damage as they grow and metastasize. We originally used HCT116 cells as a cancer model because they contain a near diploid karyotype, making them less likely to have abnormalities in gene copy number.

Reviewer #2 (Comments to the Authors (Required)):

This study provides novel insights into plasma membrane repair mechanisms by demonstrating that annexin-containing microvesicles (MVs) are secreted in response to calcium influx following membrane damage. The findings highlight the role of calpains in cleaving annexins, which facilitates release of membrane scabs and the secretion of damaged membrane and annexins as MVs.

I find the study thorough and quite compelling. The analogy between organismal wound healing, where scabs are removed after healing, and the repair of single-cell membranes where annexin scabs are removed by calpains is particularly interesting and could be emphasized more strongly in the abstract.

I only have a few comments/suggestions for improvements:

I understand that the focus here is secretion of MVs during membrane repair. However, many reports show that damaged membranes are also internalized upon injury and repair by endocytic mechanisms (e.g. upon SLO treatment) and recycled in the intracellular stomach. This aspect should be included in the text, since annexin/calpain containing scabs may also be internalized and subsequently digested in lysosomes.

Good point. Robust shedding of FM1-43-stained material occurred in all ablation experiments performed. Very rarely, (<1 in 10 movies) we also visualized some internalization of FM1-43-stained material. However, internalization may be harder to visualize, and we may be underreporting the contribution of endocytosis to membrane repair. We have included these points in our discussion.

In Figure 5i: "Addition of annexin A2 or, to a larger extent, addition of annexin A2-S100A10 caused liposomes to aggregate (Figure 5i). Addition of calpain-1 dissolved annexin A2 and annexin A2-S100A10 liposome aggregates."

Please provide some quantification of this!

Done.

Annexins are known to induce strong membrane curvature. The combination of membrane crosslinking and curvature induction around the wound site could form a strong platform for subsequent secretory or internalization events of damaged membrane (and scab). This aspect of annexin-mediated repair should also be included in the manuscript.

We completely agree. We also suspect that calpain cleavage may disrupt membrane curvature induced by annexin binding. We have included these points in the discussion.

Reviewer #3 (Comments to the Authors (Required)):

The authors discuss the role of calpain in the repair of wounded cell membranes. Following injury, calcium influx triggers the recruitment of annexins to the wound site, where they form complexes with membrane lipids to plug the wound. This initial patch is later released as microvesicles (MVs). Calpains cleave annexins, detaching them from the MV membrane, thus facilitating the release of MVs. The authors utilize advanced techniques to demonstrate the molecular interactions involved in this process. However, there are several areas requiring improvement, as listed below:

1. Page 5, Line 1: Previous experiments using pulse-laser ablation are conducted over extremely short durations (~10 ms). Longer laser exposure, such as 1 minute, may cause significant damage to cytoplasmic and membrane molecules. Furthermore, conventional

pulse-laser ablation disrupts not only the cell membrane but also the underlying actin cortex. A recent technique utilizing a gold-coated substrate can target only the cell membrane (Cells, 13, 341, 2024).

We completely agree. In fact, laser ablation has been used to disrupt actin filaments during cell division. We have included more caveats concerning the laser ablation methods used to the main text.

2. Page 5, Line 3: When does the wound pore close? The closing time might be estimated by tracking the incorporation of FM dye or SYTOX Green fluorescence.

Using both FM1-43 or SYTOX Green staining, there is an initial burst of dye influx ~1 minute in length. Afterward dye influx is much more gradual, so it is hard to pinpoint the exact transition from scabbing to sealing. We approximate 5-10 minutes.

3. Page 5, Line 4: The authors state that vesicles are generated at the wound site. However, the provided video and figures are unconvincing. Please include more definitive evidence of vesicle formation from the damaged membrane.

We have provided additional videos of the repair cap shedding process. Hopefully this satisfies the reviewer's concerns.

4. Page 5, Paragraph 4: Why do SLO pores remain open?

In that experiment, we want to measure how quickly SLO pores open, without repair of the lesion. Extracellular Ca^{2+} is required to trigger the repair response, so without Ca^{2+} in the medium, pores are not repaired and stay open after formation. This effect is well described and has been used to deliver protein complexes up to 100 kDa in length into cells (30,31).

5. Page 6, Paragraph 3: Please specify the molecular weight of each cleaved annexin in both the text and figures.

Done.

6. Page 6, Paragraph 1: Is the molecular weight of the cleaved annexins in cell lysates identical to those found in MVs? Please clarify their molecular weights.

Yes, it appears that they are the same molecular weight as assessed by immunoblot. We have clarified the molecular weights in the text.

7. Page 6, Paragraph 2: Does the calpain inhibitor prevent wound repair? Does calpain accumulate at the wound site?

We did not try this experiment, but others have shown that in mammalian cells, calpain 1/calpain 2 are required for wound repair and appear to accumulate at damage sites (Mellgren et al. *JBC* 2007).

8. *Figure 6: Please add a timeline indicating the opening and closing of the wound pore, the accumulation of annexin, calpain, and S100A10, as well as the generation and release of MVs.*

Great suggestion. We have added timings to Figure 6F.

9. *Figure 6F: This figure lacks clarity. Where does the membrane for the scab originate? Since annexins are located inside MVs, the membrane is unlikely to come from preexisting intracellular vesicles such as lysosomes or other organelles. If the membrane is derived from the wounded cell membrane, FM dye fluorescence would not increase at the wound site. It is plausible that new membrane vesicles are formed, as supported by recent findings in Dictyostelium cells.*

We believe that the scab membrane is derived from lateral compression of the plasma membrane. Indeed, laser ablation experiments indicate cell shrinking and pinching toward the wound site (Video 1-2). Endosomes and lysosomes may indirectly provide membrane by fusing distally to the wound site. We have edited Figure 6F to make it clearer.

10. *Page 10, Paragraph 2: Calpain inhibition or the expression of uncleavable annexin significantly reduces MV secretion. However, the inhibition is modest, suggesting that additional mechanisms contribute to MV secretion.*

We concur. Many other factors and pathways are implicated in membrane repair. There may be redundancies that allow shedding, independently of calpain activity. We have included this point in the discussion.

11. *Discussion: Previous studies indicate that calpains disrupt the actin cortex to promote wound repair. Please include a discussion on how the present model relates to these earlier findings.*

Great point. We do not discount previous models suggesting that calpain cleavage of the actin cortex is required for shedding and included that in the discussion.

12. *Discussion: Annexins contain annexin repeat domains that bind lipids, which persist after N-terminal cleavage by calpain. Please explain why cleaved annexins are released from the MV membrane.*

We only suspect that the short, N-terminal fragment may diffuse out of the MV. As you point out, the core annexin domain remains associated with the MV.

13. *Discussion: Is the release of wound scabs outside a common mechanism in eukaryotic cells? Please expand on this in the discussion.*

Yes, scab shedding has been documented in other eukaryotic cells including muscle cells. We expanded this point in the discussion.

14. Discussion: The same authors previously proposed that multivesicular bodies (MVBs) play a role in cellular wound repair (Williams et al., 2003, eLife). This is not mentioned in the current paper. Please include a discussion of the relationship between MVBs and MVs in wound repair.

Good point. We have incorporated more discussion of our previous paper.

15. Introduction, Line 5: Replace "intracellular" with "intercellular."

Done.

16. Introduction, Line 25: Reference number 12 should be updated to 13.

We believe that that reference is correct. Although calpains are not the main topic of that paper, they perform a targeted screen for factors required for membrane repair and one of their top hits is calpain-1.

17. Page 4, Results: Correct "tritonX-100" to "Triton X-100."

Done.

18. Page 5: Correct "1 μ M" to "1 μ m."

Done.

March 15, 2025

RE: JCB Manuscript #202408159R

Randy Schekman
University of California, Berkeley

Dear Dr. Schekman,

Thank you for submitting your revised manuscript entitled "Calpains Orchestrate Secretion of Annexin-containing Microvesicles during Membrane Repair." We would be happy to publish your paper in JCB pending final revisions necessary to meet our formatting guidelines (see details below).

A. MANUSCRIPT ORGANIZATION AND FORMATTING:

1) Text limits: Character count for Articles is < 40,000, not including spaces. Count includes title page, abstract, introduction, results, discussion, and acknowledgments. Count does not include materials and methods, figure legends, references, tables, or supplemental legends.

2) Figure formatting: Articles may have up to 10 main text figures. Scale bars must be present on all microscopy images, including inset magnifications, please add a scale bar for the inset in Fig S2B. Molecular weight or nucleic acid size markers must be included on all gel electrophoresis.

******* Size markers on gels and western blots cannot be the expected sizes of the proteins of interest. In order for readers to accurately assess the size of the proteins shown, gel and blot images must include a region containing at least one of the molecular weight size markers that were run on the gel. *******

Also, please avoid pairing red and green for images and graphs to ensure legibility for color-blind readers. If red and green are paired for images, please ensure that the particular red and green hues used in micrographs are distinctive with any of the colorblind types. If not, please modify colors accordingly or provide separate images of the individual channels.

3) Statistical analysis: Error bars on graphic representations of numerical data must be clearly described in the figure legend. The number of independent data points (n) represented in a graph must be indicated in the legend. Please, indicate whether 'n' refers to technical or biological replicates (i.e. number of analyzed cells, samples or animals, number of independent experiments). If independent experiments with multiple biological replicates have been performed, we recommend using distribution-reproducibility SuperPlots (please see Lord et al., JCB 2020) to better display the distribution of the entire dataset, and report statistics (such as means, error bars, and P values) that address the reproducibility of the findings.

Statistical methods should be explained in full in the materials and methods. For figures presenting pooled data the statistical measure should be defined in the figure legends. Please also be sure to indicate the statistical tests used in each of your experiments (both in the figure legend itself and in a separate methods section) as well as the parameters of the test (for example, if you ran a t-test, please indicate if it was one- or two-sided, etc.). Also, if you used parametric tests, please indicate if the data distribution was tested for normality (and if so, how). If not, you must state something to the effect that "Data distribution was assumed to be normal but this was not formally tested."

4) Materials and methods: Should be comprehensive and not simply reference a previous publication for details on how an experiment was performed. Please provide full descriptions (at least in brief) in the text for readers who may not have access to referenced manuscripts. The text should not refer to methods "...as previously described." Please also indicate the acquisition and quantification methods for immunoblotting/western blots.

5) For all cell lines, vectors, constructs/cDNAs, etc. - all genetic material: please include database / vendor ID (e.g. Addgene, ATCC, etc.) or if unavailable, please briefly describe their basic genetic features, even if described in other published work or gifted to you by other investigators (and provide references where appropriate). Please be sure to provide the sequences for all of your oligos: primers, si/shRNA, RNAi, gRNAs, etc. in the materials and methods. You must also indicate in the methods the source, species, and catalog numbers/vendor identifiers (where appropriate) for all of your antibodies, including secondary. If antibodies are not commercial, please add a reference citation if possible.

- 6) Microscope image acquisition: The following information must be provided about the acquisition and processing of images:
- Make and model of microscope
 - Type, magnification, and numerical aperture of the objective lenses
 - Temperature
 - Imaging medium
 - Fluorochromes
 - Camera make and model
 - Acquisition software
 - Any software used for image processing subsequent to data acquisition. Please include details and types of operations involved (e.g., type of deconvolution, 3D reconstitutions, surface or volume rendering, gamma adjustments, etc.).
- 7) References: There is no limit to the number of references cited in a manuscript. References should be cited parenthetically in the text by author and year of publication. Abbreviate the names of journals according to PubMed.
- 8) Supplemental materials: Articles may have up to 5 supplemental figures and 10 videos. Please also note that tables, like figures, should be provided as individual, editable files. A summary of all supplemental material should appear at the end of the Materials and methods section. Please include one brief sentence per item.
- 9) Video legends: Should describe what is being shown, the cell type or tissue being viewed (including relevant cell treatments, concentration and duration, or transfection), the imaging method (e.g., time-lapse epifluorescence microscopy), what each color represents, how often frames were collected, the frames/second display rate, and the number of any figure that has related video stills or images.
- 10) eTOC summary: A ~40-50 word summary that describes the context and significance of the findings for a general readership should be included on the title page. The statement should be written in the present tense and refer to the work in the third person. It should begin with "First author name(s) et al..." to match our preferred style.
- 11) Conflict of interest statement: JCB requires inclusion of a statement in the acknowledgements regarding competing financial interests. If no competing financial interests exist, please include the following statement: "The authors declare no competing financial interests." If competing interests are declared, please follow your statement of these competing interests with the following statement: "The authors declare no further competing financial interests."
- 12) A separate author contribution section is required following the Acknowledgments in all research manuscripts. All authors should be mentioned and designated by their first and middle initials and full surnames. We encourage use of the CRediT nomenclature (<https://casrai.org/credit/>).
- 13) ORCID IDs: ORCID IDs are unique identifiers allowing researchers to create a record of their various scholarly contributions in a single place. Please note that ORCID IDs are required for all authors. At resubmission of your final files, please be sure to provide your ORCID ID and those of all co-authors.
- 14) JCB requires authors to submit Source Data used to generate figures containing gels and Western blots with all revised manuscripts. This Source Data consists of fully uncropped and unprocessed images for each gel/blot displayed in the main and supplemental figures. Molecular weight or nucleic acid size markers must be included on all gel and blot Source Data images.
- Since your paper includes cropped gel and/or blot images, please be sure to provide one Source Data file for each figure. File names for Source Data figures should be alphanumeric without any spaces or special characters (i.e., SourceDataF#, where F# refers to the associated main figure number or SourceDataFS# for those associated with Supplementary figures). For traditional gels and blots, the lanes of the gels/blots should be labeled as they are in the associated figure, the place where cropping was applied should be marked (with a box), and molecular weight/size standards should be labeled. For assays performed using capillary electrophoresis and/or immunoassay-based detection, authors should instead provide the electropherogram graph(s) for each experiment, plotting fluorescence/chemiluminescence intensity vs. molecular weight/size. Each trace in the graph should be color-coded and labeled to indicate which protein, gene, or sample is being measured (please try to avoid red/green combinations to accommodate our color-blind readers).
- 15) Journal of Cell Biology now requires a data availability statement for all research article submissions. These statements will be published in the article directly above the Acknowledgments. The statement should address all data underlying the research presented in the manuscript. Please visit the JCB instructions for authors for guidelines and examples of statements at (<https://rupress.org/jcb/pages/editorial-policies#data-availability-statement>).

B. FINAL FILES:

****It is JCB policy that if requested, original data images must be made available to the editors. Failure to provide original images upon request will result in unavoidable delays in publication. Please ensure that you have access to all original data images prior to final submission.****

****The license to publish form must be signed before your manuscript can be sent to production. A link to the electronic license to publish form will be sent to the corresponding author only. Please take a moment to check your funder requirements before choosing the appropriate license.****

Thank you for your attention to these final processing requirements. Please revise and format the manuscript and upload materials within 7-14 days. If you need an extension for whatever reason, please let us know and we can work with you to determine a suitable revision period.

Thank you for this interesting contribution, we look forward to publishing your paper in Journal of Cell Biology.

Sincerely,

William Bement, PhD
Monitoring Editor
Journal of Cell Biology

Dan Simon, PhD
Scientific Editor
Journal of Cell Biology